# SELECTIVE LABELING WITH FALSE DISCOVERY RATE CONTROL

## ABSTRACT

Obtaining high-quality labels for large datasets is expensive, requiring massive annotations from human experts. While AI models offer a cost-effective alternative by predicting labels, their label quality is compromised by the unavoidable labeling errors. Existing methods mitigate this issue through selective labeling, where AI labels a subset and human labels the remainder. However, these methods lack theoretical guarantees on the quality of AI-assigned labels, often resulting in unacceptably high labeling error within the AI-labeled subset. To address this, we introduce **Conformal Labeling**, a novel method to identify subsets where AI predictions can be provably trusted. This is achieved by controlling the false discovery rate (FDR), the expected proportion of incorrect labels in the selected subset. In particular, we construct a conformal $p$-value for each test instance by comparing AI models' confidence to those of calibration instances mislabeled by AI models. Then, we select test instances whose $p$-values are below a data-dependent threshold, certifying AI models' predictions as trustworthy. We provide theoretical guarantees that Conformal Labeling controls the FDR below the nominal level, ensuring that a predefined fraction of AI-assigned labels is correct on average. Extensive experiments demonstrate that our method achieves tight FDR control with high power across various tasks, including image and text labeling, and LLM QA.

## 1 INTRODUCTION

Large-scale, high-quality labeled data is crucial for the machine learning pipelines (Deng et al., 2009). While experts could provide high-quality labels for moderately sized datasets, the growing size of modern datasets has made this approach prohibitively expensive. AI models offer a cost-effective alternative by predicting labels, bypassing the need for human experts. However, AI models are prone to labeling error (Northcutt et al., 2021; Tan et al., 2024). For example, empirical evidence demonstrates that even state-of-the-art LLMs exhibit high labeling error when used for text annotation (Baumann et al., 2025). The labeling error inherent to AI models significantly compromises their label quality, hindering the deployment of AI labeling for production. To balance the trade-off between labeling cost and error, selective labeling has been a promising solution (Geifman & El-Yaniv, 2017; Wang et al., 2023) by combining AI predictions with expert annotations.

Prior work on selective labeling primarily designed heuristic methods (Wang et al., 2021; Bernhardt et al., 2022; Wang et al., 2024a) that rely on AI models to label high-confidence instances while deferring the rest to human experts. However, these methods do not provide any theoretical guarantees on label quality. To address this, a recent work (Candès et al., 2025) proposes probably approximately correct (PAC) labeling, which guarantees that the overall labeling error is controlled with high probability. While the guarantee is theoretically appealing, the method can not control the labeling error of AI models, which can be more important in many practical applications. For instance, given a massive unlabeled dataset (e.g., LAION-5B (Schuhmann et al., 2022)), practitioners typically annotate only a subset via AI labeling, leaving the remaining instances without human verification. Additionally, one might use LLMs to synthesize extensive domain-specific texts, followed by another AI model to label a portion of them with guarantees on error rates. By iterating this synthesis-and-labeling process, it becomes feasible to build a vast annotated dataset with bounded labeling errors. Thus, it is also critical to focus reliability efforts specifically on the AI-labeled subset in some scenarios. These considerations motivate us to investigate how to provably guarantee the quality of AI-assigned labels.

In this work, we propose **Conformal Labeling**, a novel method that formulates AI-assisted labeling as a multiple-hypothesis testing problem. This method allows us to select a subset for AI labeling, providing a rigorous guarantee on the false discovery rate (FDR), i.e., the expected proportion of incorrect predictions in the selected subset. By leveraging a small labeled calibration set, we compute conformal $p$-values that measure how unusually confident a prediction is relative to known labeling errors in the calibration set. In particular, correct labels will receive small $p$-values, while incorrect ones stochastically dominate the Uniform[0,1] distribution. Then, selecting instances below a calibration-derived threshold guarantees that the FDR is at most a user-specified level, with finite-sample validity under mild assumptions. While the error rate of AI labeling in current methods depends heavily on model performance, Conformal Labeling can control the desired labeling error in expectation regardless of the underlying model's performance.

To validate our method, we conduct extensive experiments on various labeling tasks, including image labeling (ImageNet (Deng et al., 2009), ImageNet-V2 (Recht et al., 2019)), text labeling (stance on global warming (Luo et al., 2021), and misinformation (Gabriel et al., 2022)), and LLM QA (MedMCQA (Pal et al., 2022), MMLU (Hendrycks et al., 2021), MMLU-Pro (Wang et al., 2024a)) tasks. The results demonstrate that Conformal Labeling achieves high power with controlled FDR, indicating that AI models can label a large proportion of data with high quality. For example, Conformal Labeling can label 58.67% of the ImageNet dataset with ResNet-34 (He et al., 2016), while keeping the FDR below 10%. In comparison, a naive approach of using AI-assigned labels for the entire dataset results in a labeling error of over 25%. Moreover, through comprehensive ablation studies, we validate that Conformal Labeling is robust to the size of calibration datasets, and more powerful models enable better selection results.

We summarize our contributions as follows:

- We propose Conformal Labeling, a novel method for identifying subsets where AI predictions could be provably trusted. Regardless of AI models' performance, Conformal Labeling guarantees the quality of AI-assigned labels by strictly controlling the FDR.

- We theoretically prove that Conformal Labeling provides a strict quality guarantee for AI-assigned labels: it achieves valid FDR control, ensuring the expected proportion of incorrect labels is below a user-specified level.

- We empirically show that Conformal Labeling significantly reduces the labeling cost while tightly controlling the FDR, through extensive experiments conducted on image labeling, text labeling, and LLM QA tasks with various models.

## 2 PRELIMINARIES

**Problem setup.** In this work, we study the problem of identifying subsets where AI predictions can be provably trusted. Here, we give a formulation of multi-class classification as an example. Let $\mathcal{X}$ denote the feature space and $\mathcal{Y} = \{1, \ldots, K\}$ denote the label space. The test dataset $\mathcal{D}_{\text{test}} = \{X_j\}_{j=1}^m$ consists of $m$ data instances, sampled i.i.d. from a data distribution $\mathcal{P}_{\mathcal{X}}$. We denote the unseen ground-truth labels of instance $X_j$ as $Y_j$. Besides, we consider a pre-trained AI model $f : \mathcal{X} \to \mathbb{R}^{|\mathcal{Y}|}$ used to generate labels for the test dataset. For a given test instance $X$, the AI model predicts the label with the largest estimated probability $\hat{Y} = \arg\max_{y \in \mathcal{Y}} f_y(X)$, where $f_y(X)$ denotes the estimated class probability for class $y \in \mathcal{Y}$.

Since AI models unavoidably make errors when used for labeling, we aim to select a large subset from the test dataset $\mathcal{D}_{\text{label}}$ to control the portion of incorrect labels. Formally, our goal is to identify the largest subset of indices $\mathcal{R} \subseteq \{1, \cdots, m\}$ that controls the false discovery rate (FDR), defined as below:

$$\text{FDR} = \mathbb{E}\left[\frac{|\mathcal{R} \cap \mathcal{H}_0|}{\max(|\mathcal{R}|, 1)}\right], \tag{1}$$

where $\mathcal{H}_0 = \{j \in \{1, \cdots, m\} : Y_j \neq \hat{Y}_j\}$ is the set of indices with incorrect predictions and the expectation is taken over the randomness of the data. For notation shorthand, we denote $[m] = \{1, \cdots, m\}$ in the following. The FDR metric measures the expected proportion of mislabeled

samples within the AI-labeled subset, illustrating the quality of AI-assigned labels by explicitly bounding the expected fraction of incorrect labels. [1]

In addition to FDR control, we also expect AI models to label as many test instances as possible correctly, which corresponds to maximizing power:

$$\text{Power} = \mathbb{E}\left[\frac{|\mathcal{R} \cap \mathcal{H}_1|}{\max(|\mathcal{H}_1|, 1)}\right], \tag{2}$$

where $\mathcal{H}_1 = \{j \in [m] : Y_j = \hat{Y}_j\}$ is the set of indices where the AI prediction is correct. It should be emphasized that our method prioritizes FDR control over power, in that we strictly enforce FDR$\leq \alpha$ while optimizing power under the constraints.

In this work, we assume access to a small labeled calibration dataset $\mathcal{D}_{\text{cal}} = \{(X_i, Y_i)\}_{i=1}^n$. For convenience, we denote the test dataset as $\mathcal{D}_{\text{test}} = \{(X_j, Y_j)\}_{j=n+1}^{n+m}$, where $Y_j$ is not observed. Since the labeling cost of a small dataset by human annotators is typically affordable, this assumption is practical in the real world and is also adopted in prior work (Candès et al., 2025). Besides, we assume that examples of the test and calibration datasets are both drawn i.i.d. from the joint distribution $\mathcal{P}_{\mathcal{X}\mathcal{Y}}$, a common setting in selective labeling (Jung et al., 2024; Candès et al., 2025). We provide a detailed discussion of distribution shift in Appendix H.2.

**Selective labeling methods and their limitations.** Ensuring high-quality labels while reducing annotation costs has motivated extensive research on selective labeling. Prior work on selective labeling primarily focused on heuristic methods. For example, some studies design collaborative annotation frameworks that combine expert labels with LLM labels to streamline the annotation process (Li et al., 2023; Kim et al., 2024). Others propose domain-specific methods, such as meta-learning strategies for medical image labeling (Vrabac et al., 2022), annotation frameworks for text data (Duan & Lalor, 2023), and human–AI collaborative systems for object detection (Zhang et al., 2025). Although these heuristic approaches effectively reduce annotation costs, they lack formal guarantees on the label quality, which can result in unreliable labels when AI models perform poorly.

Probably approximately correct (PAC) labeling (Candès et al., 2025) addresses this limitation by providing a theoretical guarantee: the overall labeling error across the dataset is controlled with high probability. At its core, PAC labeling strategically collects zero-error expert labels for instances where the AI model exhibits the highest uncertainty, while relying on potentially imperfect AI predictions for more certain instances. This strategic allocation ensures that the dataset's overall labeling error is small, as the zero-error expert annotations effectively counterbalance the error introduced by the AI-assigned labels. While the guarantee is theoretically appealing, the method can not control the labeling error of AI models, which can be more important in many practical applications. For instance, given a massive unlabeled dataset (e.g., LAION-5B (Schuhmann et al., 2022)), practitioners typically annotate only a subset via AI labeling, leaving the remaining instances without human verification. Additionally, one might use LLMs to synthesize extensive domain-specific texts, followed by another AI model to label a portion of them with guarantees on error rates. By iterating this synthesis-and-labeling process, it becomes feasible to build a vast annotated dataset with bounded labeling errors. Thus, it is also critical to focus reliability efforts specifically on the AI-labeled subset in some scenarios. These considerations motivate us to investigate how to provably guarantee the quality of AI-assigned labels.

## 3 METHOD

### 3.1 CONFORMAL LABELING

Our previous section shows that existing methods cannot guarantee the quality of AI-assigned labels. To address this, we propose **Conformal Labeling**, which identifies subsets where AI predictions can be provably trusted by controlling the FDR. Our approach is composed of three primary steps: quantifying uncertainty, constructing conformal $p$-values, and thresholding.

**Uncertainty quantification.** Our approach builds on a key insight: we should select instances where the model exhibits high confidence in its predictions. To quantify the model confidence,

---

[1]The realized FDR might exceed the target level for a single run.

---

**Algorithm 1** Conformal Labeling

---

**Require:** Mislabeled Calibration set $\mathcal{D}_{\text{cal}}^0 = \{(X_i, Y_i)\}_{i=1}^{n_0}$, test instances $\{X_{n+j}\}_{j=1}^m$, pre-trained classifier $f$, calibration set size $n = |\mathcal{D}_{\text{cal}}|$, FDR target $\alpha \in (0, 1)$.

1: # *1. Compute uncertainty scores*
2: **for** $i = 1, \ldots, n + m$ **do**
3:     Compute $\mathcal{S}_i := 1 - \max_{y \in \mathcal{Y}} f_y(X_i)$.
4: **end for**
5: # *2. Construct conformal p-values*
6: **for** $j = 1, \ldots, m$ **do**
7:     Construct $\hat{p}_j$ according to equation 3.
8: **end for**
9: # *3. Thresholding*
10: Compute $j^* = \max\left\{j : \hat{p}_{(j)} \leq \frac{\alpha j (n+1)}{m(n_0 + 1)}\right\}$, where $\hat{p}_{(j)}$ is the $j$-th smallest p-value.
11: **Output:** $\mathcal{R} = \{j : \hat{p}_j \leq \hat{p}_{(j^*)}\}$.

---

we define an uncertainty score $\mathcal{S} : \mathcal{X} \to \mathbb{R}$, where a higher value indicates greater model uncertainty. We note that in the conformal inference framework, this score function is also known as the non-conformity score function. For example, we employ $\mathcal{S}(X) = 1 - \max_{y \in \mathcal{Y}} f_y(X)$ as our uncertainty score function (Hendrycks & Gimpel, 2016). This score is a valid measure of uncertainty, since prior works establish that misclassified samples generally receive lower probability scores (i.e., $\max_{y \in \mathcal{Y}} f_y(X)$) than correctly classified ones (Hendrycks & Gimpel, 2016; Tu et al., 2024).

**Statistical guarantee via conformal $p$-value.** To provide a statistical guarantee, we reformulate our problem as the following multiple hypothesis testing problem:

$$H_j^0 : Y_{n+j} \neq \hat{Y}_{n+j} \quad \text{v.s.} \quad H_j^1 : Y_{n+j} = \hat{Y}_{n+j}, \quad \forall j = 1, \cdots, m$$

Rejecting the null hypothesis $H_j^0$ indicates that $(X_{n+j}, \hat{Y}_{n+j})$ should be included in the subset, as it is deemed to be classified correctly. To construct the selection subset, we employ *conformal p-value*, which builds upon conformal inference framework (Vovk et al., 1999; 2005). The underlying intuition is that: *a test instance $X$ is likely a misclassification if its uncertainty score $\mathcal{S}(X)$ is generally larger than the scores of instances that are known to be misclassified.* Leveraging this idea, conformal $p$-value is computed through a rank-based comparison of $\mathcal{S}(X)$ against uncertainty scores of misclassified instances.

Formally, for the calibration dataset $\mathcal{D}_{cal}$, we identify the subset $\mathcal{D}_{cal}^0 \subseteq \mathcal{D}_{cal}$ where instances are misclassified by the AI model. For simplicity, we denote $\mathcal{D}_{cal}^0 = \{(X_i, Y_i)\}_{i=1}^{n_0}$, and thus $Y_i \neq \hat{Y}_i$ for $i = 1, \cdots, n_0$. We compute the uncertainty scores for the entire dataset $\{(X_i, Y_i)\}_{i=1}^{n+m}$: $\mathcal{S}_i = 1 - \max_{y \in \mathcal{Y}} f_y(X_i)$. Then, the conformal $p$-value for the instance $X_{n+j}$ is computed by

$$\hat{p}_j = \frac{\sum_{i=1}^{n_0} \mathbf{1}\{\mathcal{S}_i < \mathcal{S}_{n+j}\} + (1 + \sum_{i=1}^{n_0} \mathbf{1}\{\mathcal{S}_i = \mathcal{S}_{n+j}\}) \cdot U_j}{n_0 + 1}, \tag{3}$$

where $U_j \sim \text{Uniform}[0, 1]$ are i.i.d. uniform random variable to randomize over ties when $S_{n+j}$ equals some $S_i$, ensuring a continuous conformal $p$-value. Here, $\hat{p}_j$ quantifies how extreme the uncertainty of $X_{n+j}$ is compared to the scores of misclassified instances, with a small $\hat{p}_j$ providing strong statistical evidence for correct prediction.

Standard results from conformal inference establish the validity of conformal $p$-value in practice: if the instance $X_{n+j}$ is misclassified, then the conformal $p$-value stochastically dominates the uniform distribution on $[0, 1]$ (Bates et al., 2023; Jin & Candès, 2023): $\mathbb{P}\{\hat{p}_j \leq \alpha \mid H_j^0 \text{ is true}\} \leq \alpha$. This property indicates that the conformal $p$-values for misclassified instances are biased to be high, which allows us to set a threshold to flag potential correct instances, while controlling the FDR.

**Thresholding.** After obtaining conformal $p$-values for test instances, we apply a thresholding rule inspired by the Benjamini–Hochberg (BH) procedure (Benjamini & Hochberg, 1995) to select a maximal subset $\mathcal{R}$ for AI labeling while controlling the FDR at level $\alpha$. The key idea is that we gradually increase the acceptance threshold until including additional samples would risk exceeding

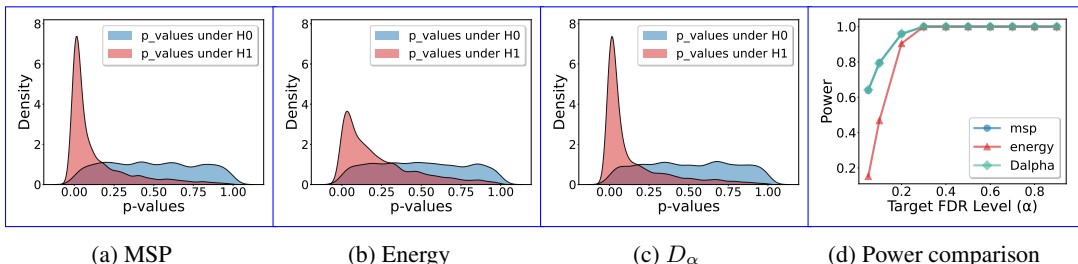

|                  |                  |                  |                  |
|:----------------:|:----------------:|:----------------:|:----------------:|
| (a) MSP          | (b) Energy       | (c) $D_\alpha$   | (d) Power comparison |

Figure 1: **Empirical distributions and power (employed with our method) of conformal $p$-values under different uncertainty scores.** The experiments include maximum softmax probability (MSP), energy, and DOCTOR-$\alpha$ score ($D_\alpha$). The experiments are conducted on ImageNet with ResNet-34. The results show that both MSP and $D_\alpha$ score create a clear distinction between correct and incorrect predictions, thus achieving high statistical power. However, the energy score fails to provide this separation, leading to low power.

the desired labeling error. In particular, let $p_{(1)} \leq \ldots \leq p_{(m)}$ denote the ordered statistics of the $p$-values; the rejection set of our selection procedure applied to the conformal $p$-values is $\mathcal{R} = \{j : \hat{p}_j \leq \hat{p}_{(j^*)}\}$, where

$$j^* = \max\left\{ j : \hat{p}_{(j)} \leq \frac{\alpha j(n+1)}{m(n_0+1)} \right\},$$

with the convention that $\max\varnothing = 0$. We summarize the complete procedure of Conformal Labeling in Algorithm 1, which combines all three steps described above.

**Theoretical results.** We now provide a theoretical guarantee for Conformal Labeling. In the following theorem, we establish that Algorithm 1 controls the FDR below the desired level $\alpha$. This theorem ensures that AI predictions can be provably trusted in the selected subset, as the expected proportion of incorrect labels is controlled below $\alpha$. The proof is provided in Appendix B.2.

**Theorem 3.1.** *Suppose calibration samples $(X_i, Y_i)_{i=1}^n$ and test samples $(X_{n+j}, Y_{n+j})_{j=1}^m$ are i.i.d. Let $\alpha \in (0, 1)$ be the target FDR level, and suppose the selection set $\mathcal{R}$ is determined by Algorithm 1 applied at the target FDR level $\alpha$. Define $p = \mathbb{E}[H_j^0]$, the probability that a test sample $(X_{n+j}, Y_{n+j})$ is incorrectly predicted. Taking expectation over the randomness of the calibration and test data, the FDR of the selection set $\mathcal{R}$ satisfies:*

$$\text{FDR} \leq \left[ 1 - (1-p)^{n+1} \right] \alpha \leq \alpha.$$

### 3.2 CHOICE OF UNCERTAINTY SCORE

In the above analysis, we establish that Conformal Labeling controls the FDR below the desired level. However, this guarantee alone is insufficient: a trivial procedure that simply labels nothing would achieve a perfect FDR of 0, yet offer no practical value. This highlights the need to also evaluate the method's statistical power, which measures the method's ability to identify as many correctly labeled instances as possible (see Eq. (2)).

As shown in prior work (Jin & Candès, 2023; Gui et al., 2024; Bai et al., 2025), the statistical power of this method depends on the quality of the uncertainty score. In particular, a score that better separates correct from incorrect predictions directly increases statistical power (see Proposition 7 of Jin & Candès (2023)). To deliver practical recommendations, we empirically compare several uncertainty scores by visualizing their resulting $p$-value distributions and measuring the final statistical power employed with our method. We utilize a pre-trained ResNet-34 model on the ImageNet dataset, with three uncertainty scores: maximum softmax probability (MSP) (Hendrycks & Gimpel, 2016), energy score (Liu et al., 2020), and DOCTOR-$\alpha$ score ($D_\alpha$) (Granese et al., 2021). We give an overview of these score functions in Appendix C. The results in Figure 1 show that both MSP and $D_\alpha$ score provide a clear distinction between correct and incorrect predictions, thus achieving high statistical power. However, the energy score fails to provide this separation, leading to low power. Given the comparable power performance of MSP and $D_\alpha$ score, we will use the more computationally efficient MSP in our main experiments. In Appendix I.2, we give a more detailed comparison of the power and FDR achieved by these score functions.

Table 1: **Performance of Conformal Labeling on three labeling tasks.** $\uparrow$ indicates that a larger value is better. We evaluate on Image Labeling (ImageNet, ImageNet-V2), text Labeling (Stance on global warming, Misinformation), and LLM QA (MedMCQA, MMLU) tasks. We report results for Conformal Labeling at $\alpha = 0.1$ and compare Conformal Labeling against two baselines: (i) a naive approach of labeling instances whose uncertainty score $\mathcal{S}_{n+j} \leq 0.1$ with AI models, and (ii) labeling the entire dataset with AI models. The results show that Conformal Labeling consistently achieves tighter FDR control across all datasets and models compared with the baseline.

| Task | Dataset | Model | Conformal Labeling ($\alpha$= 0.1) | | | Naive ($\mathcal{S} \leq 0.1$) | | | AI only |
|------|---------|-------|-------|-------------|-------------|-------|-------------|-------------|---------|
| | | | FDR % | Power %($\uparrow$) | Ratio %($\uparrow$) | FDR % | Power %($\uparrow$) | Ratio %($\uparrow$) | Error %($\downarrow$) |
| Image | ImageNet | ResNet-34 | 9.97 | 80.01 | 58.67 | 4.79 | 63.09 | 43.71 | 26.71 |
| | | DenseNet-161 | 9.99 | 85.03 | 65.56 | 5.58 | 72.08 | 52.98 | 22.89 |
| | | ResNeXt50 | 10.00 | 86.06 | 66.83 | 6.08 | 75.57 | 56.21 | 22.39 |
| | | CLIP-VIT-B/32 | 9.98 | 46.04 | 27.47 | 5.53 | 28.65 | 16.28 | 40.35 |
| | ImageNet-V2 | ResNet-34 | 10.00 | 61.99 | 37.87 | 7.25 | 56.03 | 33.15 | 39.03 |
| | | DenseNet-161 | 9.83 | 66.67 | 43.39 | 9.39 | 65.63 | 42.45 | 34.88 |
| | | ResNeXt50 | 9.95 | 67.86 | 44.75 | 10.17 | 68.71 | 45.38 | 34.08 |
| | | CLIP-VIT-B/32 | 9.96 | 34.56 | 18.10 | 7.86 | 25.83 | 13.18 | 47.78 |
| Text | Stance | Llama-3.1-8B-Instruct | 9.77 | 19.02 | 9.42 | 24.43 | 54.94 | 31.39 | 52.04 |
| | | Qwen3-32B | 9.56 | 10.70 | 7.25 | 14.42 | 16.64 | 11.11 | 36.52 |
| | | Qwen2.5-72B-Instruct | 9.71 | 26.97 | 17.84 | 26.27 | 74.49 | 59.01 | 35.09 |
| | Misinformation | Llama-3.1-8B-Instruct | 9.88 | 7.31 | 5.81 | 18.38 | 66.17 | 55.25 | 24.28 |
| | | Qwen3-32B | 9.91 | 49.21 | 37.48 | 10.77 | 52.27 | 40.03 | 24.08 |
| | | Qwen2.5-72B-Instruct | 9.58 | 44.36 | 34.81 | 17.81 | 87.13 | 74.72 | 21.69 |
| QA | MedMCQA | Llama-3.1-8B-Instruct | 9.70 | 31.44 | 18.90 | 15.01 | 46.74 | 29.53 | 40.35 |
| | | Qwen3-32B | 9.75 | 49.80 | 33.27 | 13.79 | 65.27 | 45.36 | 33.44 |
| | | Llama-3.1-70B-Instruct | 9.95 | 69.67 | 49.59 | 4.52 | 48.92 | 32.79 | 28.90 |
| | MMLU | Llama-3.1-8B-Instruct | 9.99 | 58.25 | 37.47 | 8.47 | 53.72 | 33.96 | 35.72 |
| | | Qwen3-32B | 10.00 | 82.96 | 65.22 | 8.13 | 78.47 | 60.40 | 21.43 |
| | | Llama-3.1-70B-Instruct | 9.96 | 88.20 | 72.10 | 4.18 | 67.94 | 52.17 | 18.24 |

## 4 EXPERIMENTS

In this section, we evaluate the effectiveness of Conformal Labeling on image labeling, text labeling, and LLM QA tasks with various models. We find that it achieves tight FDR control and high power, indicating that AI models can label a large proportion of data with high quality. We also conduct comprehensive ablation studies to provide practical guidance for applying our method.

### 4.1 EXPERIMENTAL SETUP

**Tasks and datasets.** We evaluate the effectiveness of Conformal Labeling across three labeling tasks, including image labeling, text labeling, and LLM question answering (QA). In Appendix G, we also demonstrate how to extend Conformal Labeling to regression tasks. For the LLM QA task, the goal is to identify subsets of questions that LLMs can answer correctly. We employ common benchmark datasets for evaluations in each labeling task. For image classification, we use ImageNet (Deng et al., 2009) and its variant ImageNet-V2 datasets (Recht et al., 2019). For text labeling, we adopt two benchmark datasets. The first is Stance on Global Warming (Luo et al., 2021), which provides annotations ($Y_i \in \{\text{agree}, \text{neutral}, \text{disagree}\}$) to judge whether a headline agrees that global warming is a serious concern. The second is Misinformation (Gabriel et al., 2022), which contains binary annotations ($Y_i \in \{\text{misinfo}, \text{real}\}$) for identifying whether a given text contains misinformation. For the LLM QA task, we evaluate our method on MedMCQA (Pal et al., 2022), MMLU (Hendrycks et al., 2021), and MMLU-Pro (Wang et al., 2024b) datasets.

**Models.** We conduct extensive experiments on various open-sourced AI models. For image classification, we utilize three well-established deep image classifiers: ResNet-34 (He et al., 2016), DenseNet-161 (Huang et al., 2018), and ResNext50 (Xie et al., 2017). Additionally, we employ the Vision-Language Model CLIP (Radford et al., 2021), which is based on a Vision Transformer architecture (ViT-B/32) (Dosovitskiy et al., 2021). The above classifiers are provided by TorchVi-

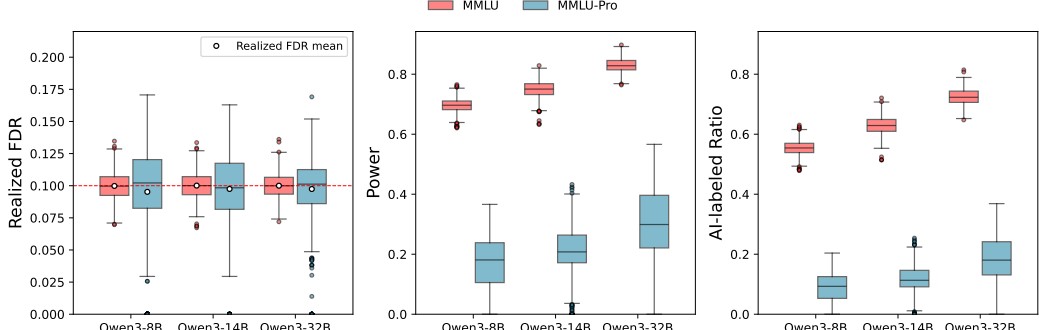

Figure 2: **Performance comparison of Conformal Labeling across models of varying accuracy.** We employ Qwen3-8B, Qwen3-14B, and Qwen3-32B (model accuracy increases with parameter count) on MMLU and MMLU-Pro with $\alpha = 0.1$. The results show that model with higher accuracy achieves greater power and AI-labeled ratio.

sion (Paszke et al., 2019). For text labeling, we employ three LLMs: Llama-3.1-8B-Instruct (Dubey et al., 2024), Qwen3-32B, Qwen2.5-72B-Instruct (Qwen et al., 2025). For LLM QA tasks, we employ five LLMs: Qwen3-8B (Yang et al., 2025), Qwen3-14B, Qwen3-32B, Llama-3.1-8B-Instruct, and Llama-3.1-70B-Instruct. The above LLMs are provided by Hugging Face.

**Baselines and evaluation metrics.** We evaluate Conformal Labeling with two baseline methods: using AI models to label test instances with uncertainty scores $\mathcal{S}_{n+j} \leq 0.1$, and applying AI predictions to the entire test dataset. We compare Conformal Labeling's selection procedure against BH (Benjamini & Hochberg, 1995), Storey-BH (Storey, 2002), and Quantile-BH procedures (Benjamini et al., 2006). We evaluate the performance of our method and baseline using the following metrics: (1) FDR, the expected proportion of incorrect labels in the selected set $\mathcal{R}$; (2) Power, the proportion of correctly labeled instances selected; (3) AI-labeled ratio, the number of data labeled by the AI models divided by the combined size of the calibration and test datasets.

**Implementation details.** To ensure the reliability of our results, we repeat each experiment 1000 times and report the average result. We randomly select 10% of the data as the calibration set. For all the experiments, we use the maximum softmax probability (MSP) as the uncertainty score function. In the LLM QA task, we adopt the standard multiple-choice evaluation pipeline: given a question and candidate answers, the model estimates the probability of each option by extracting the logits corresponding to the option tokens and applying a softmax transformation, with the predicted label taken as the option with the highest probability. For text labeling, we reformulate each sample into a multiple-choice format (e.g., "positive" or "negative"), enabling the same probability-extraction procedure as in the LLM QA task. More details of implementation are provided in Appendix F. We provide the code for reproducing our main experiments in this anonymous repository.

## 4.2 EXPERIMENTS RESULTS

**Conformal Labeling achieves tight FDR control with high power.** In table 1, we present the performance of Conformal Labeling against two baselines on three different labeling tasks: image labeling, text labeling, and LLM QA. A salient observation is that across all the labeling tasks and all the model architectures, Conformal Labeling successfully controls the FDR at or below the target FDR level. In comparison, both baseline methods lack FDR control, resulting in substantial labeling errors that compromise label quality when AI models are inaccurate. It is worth noting that the FDR is tightly controlled: for $\alpha = 10\%$, most experiments yield FDRs below 9.9%, with the largest deviation at 9.56%. This tight FDR control directly leads to high selection power. For example, on MMLU with Qwen3-32B at $\alpha = 10\%$, Conformal Labeling achieves a power of 82.96%, labeling 65.22% of the dataset with the AI model. Overall, empirical results show that Conformal Labeling consistently achieves tight FDR control with high power across different datasets and models.

Table 2: **Comparison of different selection procedures on three labeling tasks.** ↑ indicates that a larger value is better. **Bold** indicates the highest power for each row. We compare Conformal Labeling's selection procedure against three baselines: (i) BH procedure, (ii) Storey-BH procedure and (iii) Quantile-BH procedure. We report results at $\alpha = 0.1$ for all four selection procedures. The results show that Conformal Labeling's selection procedure consistently achieves tighter FDR control across all datasets and models compared with the baseline.

| Task | Dataset | Model | Conformal Labeling | | BH | | Storey-BH | | Quantile BH | |
|------|---------|-------|------|------|------|------|------|------|------|------|
| | | | FDR % | Power % | FDR % | Power %(↑) | FDR % | Power %(↑) | FDR % | Power %(↑) |
| Image | ImageNet | ResNet-34 | 9.97 | **80.01** | 2.63 | 46.85 | 8.64 | 74.57 | 7.57 | 71.79 |
| | | DenseNet-161 | 9.99 | **85.03** | 2.27 | 44.55 | 8.34 | 81.17 | 7.33 | 78.40 |
| | | ResNeXt50 | 10.00 | **86.06** | 2.19 | 47.69 | 8.17 | 81.66 | 7.22 | 79.12 |
| | | CLIP-VIT-B/32 | 9.98 | **46.04** | 4.08 | 21.70 | 8.52 | 41.02 | 8.05 | 39.23 |
| | ImageNet-V2 | ResNet-34 | 10.00 | **61.99** | 3.84 | 37.60 | 9.65 | 59.84 | 8.53 | 57.35 |
| | | DenseNet-161 | 9.83 | **66.67** | 3.39 | 28.39 | 9.62 | 65.45 | 8.44 | 61.66 |
| | | ResNeXt50 | 9.95 | **67.86** | 3.20 | 33.28 | 9.22 | 65.17 | 8.78 | 64.00 |
| | | CLIP-VIT-B/32 | 9.96 | **34.56** | 4.85 | 14.43 | 9.20 | 32.25 | 8.35 | 28.52 |
| Text | Stance | Llama-3.1-8B-Instruct | 9.77 | **19.02** | 5.23 | 8.48 | 9.23 | 17.62 | 7.42 | 16.07 |
| | | Qwen3-32B | 9.56 | **10.70** | 2.97 | 0.83 | 5.97 | 5.90 | 6.64 | 3.68 |
| | | Qwen2.5-72B-Instruct | 9.71 | **26.97** | 2.62 | 5.34 | 5.94 | 17.09 | 5.89 | 13.12 |
| | Misinformation | Llama-3.1-8B-Instruct | 9.88 | **7.31** | 1.14 | 0.06 | 4.75 | 1.73 | 4.04 | 1.37 |
| | | Qwen3-32B | 9.91 | **49.21** | 2.02 | 12.16 | 5.70 | 30.36 | 4.53 | 24.96 |
| | | Qwen2.5-72B-Instruct | 9.58 | **44.36** | 3.03 | 5.77 | 5.27 | 24.37 | 4.55 | 21.71 |
| QA | MedMCQA | Llama-3.1-8B-Instruct | 9.70 | **31.44** | 3.68 | 12.89 | 7.43 | 25.37 | 6.33 | 22.11 |
| | | Qwen3-32B | 9.75 | **49.80** | 2.71 | 10.99 | 6.99 | 34.89 | 6.34 | 31.27 |
| | | Llama-3.1-70B-Instruct | 9.95 | **69.67** | 2.79 | 29.67 | 8.48 | 64.09 | 6.74 | 58.06 |
| | MMLU | Llama-3.1-8B-Instruct | 9.99 | **58.25** | 3.40 | 29.49 | 7.31 | 49.74 | 7.17 | 49.16 |
| | | Qwen3-32B | 10.00 | **82.96** | 1.37 | 10.51 | 7.33 | 74.57 | 6.25 | 70.37 |
| | | Llama-3.1-70B-Instruct | 9.96 | **88.20** | 1.61 | 18.74 | 6.95 | 80.08 | 5.96 | 76.77 |

**Conformal Labeling outperforms existing selection procedures.** In Table 2, we present a comprehensive comparison of Conformal Labeling's selection procedure against three baselines: Storey-BH (Storey, 2002), and Quantile-BH (Benjamini et al., 2006) procedures (see Appendix D for details). We set the hyperparameters for the Storey-BH and Quantile-BH procedures with a bootstrap method detailed in Appendix E. The results show that our method can achieve much higher power than those selection procedures, with more precise FDR control. For instance, on the ImageNet dataset with ResNet-34, Conformal Labeling achieves a power of 80.01% at $\alpha = 0.1$, outperforming BH (46.85%), Storey-BH (74.57%), and Quantile-BH (71.79%) procedures. This high power is consistently observed across other labeling tasks. Overall, Conformal Labeling consistently achieves higher power than existing selection procedures across different labeling tasks.

**Higher prediction accuracy enables better selection results.** The performance of Conformal Labeling depends heavily on the underlying prediction accuracy, which is influenced by model capacity and dataset difficulty. We evaluate Conformal Labeling with Qwen3-8B, Qwen3-14B, and Qwen3-32B, whose increasing scales provide greater capacity. MMLU-Pro, more challenging than MMLU, results in lower prediction accuracy. Figure 2 presents the evaluation results. In all cases, the realized FDR remains below and close to $\alpha = 0.1$. Higher accuracy—achieved with stronger models or easier datasets—boosts power and the AI-labeled ratio. Specifically, for any given model, performance is superior on MMLU compared to MMLU-Pro. Similarly, for each dataset, larger models yield greater power and AI-labeled ratios. Overall, higher prediction accuracy leads to better selection results by achieving higher power and AI-labeled ratio with controlled FDR.

**How many calibration samples are needed?** The size of the calibration set plays a crucial role in constructing reliable conformal $p$-values. To study the effect of $|D_{cal}|$ on FDR and power, in Figure 3, we label $\{5\%, 10\%, 20\%\}$ of the unlabeled dataset as the calibration dataset. Our results demonstrate that Conformal Labeling is robust to calibration set size: even with a 5% calibration ratio, the FDR remains controlled with low standard deviation. Increasing the calibration ratio reduces the

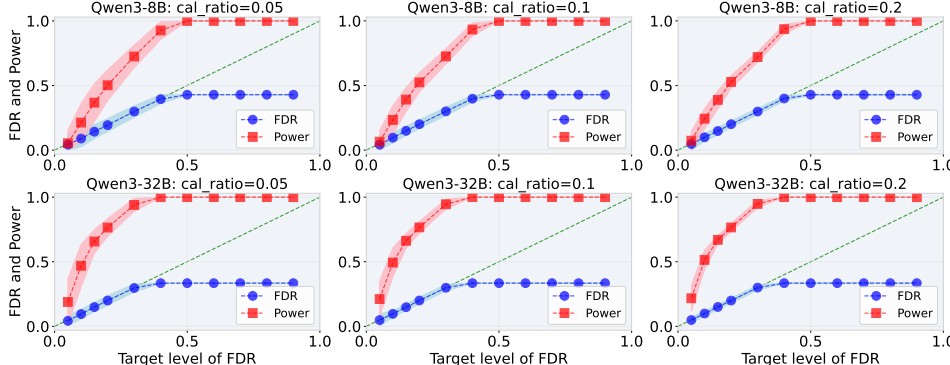

Figure 3: **Performance of Conformal Labeling with varying calibration set sizes on the MedM-CQA dataset.** The top row corresponds to the results from Qwen3-8B and the bottom row to those from Qwen3-32B; each column corresponds to a value of calibration ratio. Shaded regions indicate one standard deviation around the average. The results show that a large calibration set slightly reduces the variance of FDR and power, leading to a more robust selection outcome. Overall, our method is robust to changes in the calibration set size.

variance of both FDR and power, although the improvement from 10% to 20% is negligible. Based on this trade-off between variance reduction and labeling cost, we use a 10% calibration ratio for all the experiments. In summary, while Conformal Labeling is robust to calibration set sizes, larger calibration sets reduce the variance of FDR and power, thereby enhancing selection stability.

## 5 RELATED WORK

**Conformal Inference.**   Conformal inference (Vovk et al., 2005; Gazin et al., 2025; Humbert et al., 2025) has been extensively studied for its distribution-free and model-agnostic properties. Existing approaches can be broadly categorized into two main directions: conformal novelty detection (Bates et al., 2023; Bashari et al., 2023; Wu et al., 2025; Bashari et al., 2025; Lee et al., 2025) and conformal selection (Jin & Candès, 2023; Gui et al., 2024). The former focuses on identifying out-of-distribution instances, with recent advances focusing on enhancing selection power by incorporating various forms of side information (Liang et al., 2024; Marandon et al., 2024; Zhao & Sun, 2025). The latter aims to select candidates whose unobserved outcomes exceed user-specified values (Jin & Candès, 2023), which has been extended to multivariate data selection (Bai et al., 2025), online data selection (Xu & Ramdas, 2024; Liu et al., 2025), and human-in-the-loop adaptive data selection (Gui et al., 2025). Besides, some works have developed conformal inference techniques for classification tasks (Zhao & Su, 2023; Sun & Xia, 2025). For example, recent work (Zhao & Su, 2023) proposes controlling the overall error rate in binary and multi-class classification, while a concurrent work (Sun & Xia, 2025) extends this to control the general group-wise false discovery rate within a unified framework. In this work, we motivate and tailor the framework of conformal selection for selective labeling.

**Selective labeling and prediction.**   Selective labeling is an increasingly important topic for balancing the tradeoff between labeling cost and error (Gu et al., 2012; Vrabac et al., 2022). Prior work mainly explored heuristic approaches (Li et al., 2023; Duan & Lalor, 2023), with methods developed for the text (Kim et al., 2024), vision (Zhang et al., 2025), and medical domains (Vrabac et al., 2022). Recent works use selective labeling to enable more powerful statistical inference (Zrnic & Candes, 2024; Gligorić et al., 2024) and to construct labeled datasets with guaranteed low overall labeling error (Candès et al., 2025). In this work, we control the AI-labeled error rather than the overall labeling error. Our method is also related to selective prediction, where a model is allowed to abstain from making predictions when uncertain (El-Yaniv et al., 2010; Mozannar & Sontag, 2020; Yang et al., 2023; Kamath et al., 2020; Yoshikawa & Okazaki, 2023). However, most of these methods cannot provide theoretical guarantees on prediction error. While works on selective prediction with risk control (El-Yaniv & Wiener, 2010; Geifman & El-Yaniv, 2017) can provide theoretical guaran-

tees on prediction error, they target a different error measure. We provide a detailed comparison of the error measures and methods as discussed in Appendix H.1.

## 6 CONCLUSION

In this work, we propose Conformal Labeling, a novel selective labeling method for identifying subsets where AI predictions can be provably trusted. This is achieved by controlling the false discovery rate (FDR), which ensures that the expected fraction of incorrect labels in the selected subset is below a user-specified level. The key idea is to reformulate selective labeling as multiple hypothesis testing, which enables distinct theoretical guarantees and methodological advantages compared to prior approaches. In particular, we construct a conformal $p$-value for each test instance by comparing the AI model's predicted confidence to those of mislabeled calibration instances. Then, we select all the test samples whose $p$-values are smaller than or equal to a data-dependent threshold. We theoretically prove that Conformal Labeling successfully controls the FDR under mild assumptions. Extensive experiments demonstrate that Conformal Labeling achieves tight FDR control and high power across various tasks, including image and text labeling, and LLM QA. We hope the insight from this work will inspire future research to explore reliable selective labeling.

## 7 ETHICS STATEMENT

In this work, we study how to identify instances where AI predictions can be provably trusted in selective labeling. As our contribution focuses on enhancing the reliability of the selective labeling process, we do not see any specific ethical concerns arising from this study.

## 8 REPRODUCIBILITY STATEMENT

We have made every effort to ensure that the results presented in this paper are reproducible. We have provided code and data in this anonymous repository to allow for the verification of our main experiments. The experimental setup, including datasets, models, and hyperparameter settings, is described in detail in the paper. We have also provided a full description of the prompts, software details, and hardware details in Appendix F to assist others in reproducing our experiments. Additionally, all the datasets we use, such as ImageNet and MMLU, are publicly available, ensuring consistent and reproducible evaluation results. We believe these measures will enable other researchers to reproduce our work and further advance the field.

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

# A  USE OF LARGE LANGUAGE MODEL

This paper uses large language models solely to polish specific sentences or paragraphs, without further use of LLMs for other purposes.

# B  TECHNICAL PROOFS

## B.1  LEMMAS FOR PROVING THEOREM 3.1

**Lemma B.1.** *(Benjamini & Yekutieli, 2001, Theorem 1.2) If the joint distribution of the p-values is PRDS on the subset corresponding to true null hypotheses, applying the Benjamini–Hochberg procedure at level $\alpha$ guarantees*

$$FDR \leq \frac{m_0}{m}\alpha,$$

*where $m_0$ is the number of true null hypotheses.*

**Lemma B.2.** *Under the assumptions of Theorem 3.1, where the calibration samples $(X_i, Y_i)_{i=1}^n$ and the test samples $(X_{n+j}, Y_{n+j})_{j=1}^m$ are independently and identically distributed (i.i.d.), and conditional on $n_0$ (the number of true null hypotheses in the calibration set) and $m_0$ (the number of true null hypotheses in the test set), the expected false discovery proportion (FDP) satisfies*

$$\mathbb{E}[FDP \mid n_0, m_0] \leq \frac{1+n}{1+n_0}\frac{m_0}{m}\alpha,$$

*where $\alpha \in (0, 1)$ is the target false discovery rate (FDR) level, and the selection set $\mathcal{R}$ is determined by Algorithm 1.*

*Proof.* Without loss of generality, assume that the first $n_0$ calibration samples and the first $m_0$ test samples correspond to true null hypotheses. By the standard result in conformal inference (Vovk et al., 2005) we have,

$$P(\hat{p}_{n+j} \leq t \mid n_0, m_0) \leq t \quad \text{for all } t \in [0,1],\ j = 1, \ldots, m_0.$$

From results in Bates et al. (2023), we know conformal $p$-values $(\hat{p}_{n+1}, \ldots, \hat{p}_{n+m})$ are PRDS on the set of mislabeled data. Lemma B.1 hence guarantees that conditional on $n_0$ and $m_0$, applying Conformal Labeling at level $\alpha$ satisfies

$$\mathbb{E}[\text{FDP} \mid n_0, m_0] \leq \frac{1+n}{1+n_0}\frac{m_0}{m}\alpha,$$

$\square$

## B.2  PROOF OF THEOREM 3.1

*Proof of theorem 3.1.* Under the i.i.d. assumption of Theorem 3.1, the calibration samples $(X_i, Y_i)_{i=1}^n$ and the test samples $(X_{n+j}, Y_{n+j})_{j=1}^m$ are independently and identically distributed. This implies that each hypothesis, whether from the calibration or test set, has an equal probability $p$ of being a true null, where $p$ represents the expected probability of incorrect prediction under the null hypothesis.

Consequently, the number of true null hypotheses in the calibration set, denoted $n_0$, follows a binomial distribution $n_0 \sim \text{Binomial}(n, p)$, and the number of true null hypotheses in the test set, denoted $m_0$, follows $m_0 \sim \text{Binomial}(m, p)$. The independence across the calibration and test sets arises from the i.i.d. structure, ensuring that $n_0$ and $m_0$ are independent random variables.

Using the law of total expectation, we express FDR as

$$
\begin{aligned}
\text{FDR} &= \mathbb{E}[\text{FDP}] \\
&= \mathbb{E}[\mathbb{E}[\text{FDP} \mid n_0, m_0]] \\
&\leq \mathbb{E}\left[\frac{m_0}{m}\frac{1+n}{1+n_0}\alpha\right] \quad \text{by Lemma B.2} \\
&= \mathbb{E}\left[\frac{m_0}{m}\right] \cdot \mathbb{E}\left[\frac{1+n}{1+n_0}\right] \cdot \alpha \quad \text{since } n_0 \text{ and } m_0 \text{ are independent} \\
&= p \cdot \mathbb{E}\left[\frac{1+n}{1+n_0}\right] \cdot \alpha \quad \text{since } m_0 \sim \text{Binomial}(m, p)
\end{aligned}
\tag{4}
$$

Now it suffices to show that $p \cdot \mathbb{E}\left[\frac{1+n}{1+n_0}\right] = [1-(1-p)^{n+1}]$.

Since $n_0$ has probability mass function

$$
\mathbb{P}(n_0 = k) = \binom{n}{k}p^k(1-p)^{n-k}, \quad k = 0, 1, \ldots, n,
$$

we compute

$$
\begin{aligned}
p \cdot \mathbb{E}\left[\frac{1+n}{1+k}\right] &= p \cdot \sum_{k=0}^{n}\frac{1+n}{1+k}\binom{n}{k}p^k(1-p)^{n-k} \\
&= \sum_{k=0}^{n}\frac{(n+1)!}{(k+1)!(n-k)!}\,p^{k+1}(1-p)^{n-k} \\
&= \sum_{k=0}^{n}\binom{n+1}{k+1}p^{k+1}(1-p)^{n-k} \\
&= \sum_{l=1}^{n+1}\binom{n+1}{l}p^j(1-p)^{n+1-l} \quad \text{by letting } l = k+1 \\
&= \sum_{l=0}^{n+1}\binom{n+1}{l}p^j(1-p)^{n+1-l} - \binom{n+1}{0}p^0(1-p)^{n+1-0} \\
&= \sum_{l=0}^{n+1}\mathbb{P}(X = l) - (1-p)^{n+1} \quad \text{where } X \sim \text{Binomial}(n+1, p) \\
&= 1 - (1-p)^{n+1}
\end{aligned}
\tag{5}
$$

This completes the proof, establishing the desired bound on the FDR.

$\square$

## C  OVERVIEW OF DIFFERENT UNCERTAINTY SCORE FUNCTIONS

In this section, we provide an overview of three representative uncertainty score functions that are widely used in misclassification detection: Maximum Softmax Probability (MSP) (Hendrycks & Gimpel, 2016), the energy-based score (Liu et al., 2020), and the DOCTOR-$\alpha$ score (Granese et al., 2021). Each of these functions captures predictive uncertainty from a different perspective.

**Maximum Softmax Probability (MSP).**  The Maximum Softmax Probability (MSP) baseline (Hendrycks & Gimpel, 2016) proposes to use confidence of the AI model as an uncertainty score.

$$
S_{\text{MSP}}(x) = 1 - \max_{y \in \mathcal{Y}} p_y(x),
$$

where $p_y(x)$ is the softmax probability assigned to class $y$ for input $x$. The key idea is straightforward: if the model assigns a high probability to its most likely class, the prediction is considered

confident. Although MSP is simple and effective, it has notable limitations: it only reflects the confidence in the top-1 prediction and ignores the structure of the remaining probability distribution, which may contain useful information about uncertainty.

**Energy-based score.** The energy-based score (Liu et al., 2020) is defined as

$$S_{\text{Energy}}(x) = \log \sum_{y \in \mathcal{Y}} \exp(f_y(x)),$$

where $f_y(x)$ denotes the logit value for class $y$ and $T > 0$ is a temperature parameter. This score is derived from the concept of energy in statistical physics and leverages the log-sum-exp operator over all logits. Unlike MSP, which only considers the maximum probability, the energy score integrates information from the full logit vector, thereby providing a smoother and more informative confidence measure.

**DOCTOR-$\alpha$ score.** The DOCTOR-$\alpha$ score (Granese et al., 2021) is defined as

$$S_\alpha(x) = \sum_{y \in \mathcal{Y}} p_y(x)^2,$$

where $p_y(x)$ denotes the softmax probability for class $y$. This score is inspired by information-theoretic measures of uncertainty, as it is closely related to the quadratic Rényi entropy. The intuition is that if the predictive distribution is sharp (i.e., one class has probability close to one), then $\sum_y p_y(x)^2$ will be large, indicating high confidence. Conversely, if the distribution is flat (i.e., the model is uncertain and spreads probability mass across many classes), then $S_\alpha(x)$ will be small. Compared to MSP, the DOCTOR-$\alpha$ score leverages information from the entire probability distribution rather than only the top prediction, making it a richer measure of uncertainty for misclassification detection.

## D  BH PROCEDURE AND ITS ADAPTIVE VARIANTS

Consider testing $m$ null hypotheses $H_0^1, \ldots, H_0^m$ based on their corresponding $p$-values $\{p_1, p_2, \ldots, p_m\}$. For a true null hypothesis $H_0^j$, the corresponding $p$-value $p_j$ is a random variable that is super-uniform on $[0, 1]$ under the null hypothesis. Formally, for any $u \in [0, 1]$,

$$\mathbb{P}(p_j \leq u \mid H_0^j \text{ is true}) \leq u.$$

Define $p_{(j)}$ as the $j$-th smallest $p$-value among a set of $p$-values $\{p_1, p_2, \ldots, p_m\}$. Given a set of $p$-values $\{p_1, p_2, \ldots, p_m\}$, the BH algorithm returns $S = \{j \in \{0, \ldots, m\} : p_j \leq \frac{\alpha j^*}{m}\}$, where $\alpha$ is the target FDR level and

$$j^* = \max\{j \in \{1, \ldots, m\} : p_{(j)} \leq \frac{\alpha j}{m}\}$$

When the null $p$-values $\{p_j : j \in \mathcal{H}_0\}$ are independent, the BH procedure is proved to control the FDR at level $\pi_0 \alpha$ in finite samples (Benjamini & Hochberg, 1995), where $\pi_0 = \frac{|\mathcal{H}_0|}{m}$ is the proportion of true nulls. The independence assumption can be further relaxed to the PRDS condition (Benjamini & Yekutieli, 2001).

If $\pi_0$ is small, the FDR control will be overly conservative. For example, if a classifier achieves a 80% accuracy in the unlabeled test dataset, then $\pi_0 = 0.2$—which leads to overly conservative FDR control. When $\pi_0$ is known, we can apply BH procedure at level $\frac{\alpha}{\pi_0}$ to close the gap. In practice, $\pi_0$ is typically unknown. Several adaptive BH procedures attempt to address this issue by estimating $\pi_0$ and adjusting the target FDR level $\alpha$ accordingly. These procedures are often called the $\pi_0$-adaptive versions of the BH algorithm. Two most famous estimators are Storey-BH (Storey, 2002) and Quantile-BH (Benjamini et al., 2006):

$$\hat{\pi}_0^{Storey}(\lambda) = \frac{1 + \sum_{i=1}^m \mathbf{1}\{p_i \geq \lambda\}}{m(1 - \lambda)}, \lambda \in (0, 1)$$

$$\hat{\pi}_0^{Quant}(k_0) = \frac{m - k_0 + 1}{m(1 - p_{(k_0)})}, k_0 \in \{1, \ldots, m\}$$

.

$\lambda$ and $k_0$ are hyperparameters determined by users.

## E   HYPERPARAMETER SELECTION FOR BH ADAPTIVE VARIANTS

Both Storey-BH and Quant-BH require careful hyperparameter selection—$\lambda$ for Storey-BH and $k_0$ for Quant-BH—as this choice significantly impacts their performance. Following Storey (2002), we employ a bootstrap-based method to select the optimal $\lambda$ (and analogously, $k_0$ for the Quantile BH procedure). Further details can be found in Section 9 of Storey (2002). The algorithm proceeds as follows:

1. Define a grid $R$ for the hyperparameter, i.e. $R = \{0.1, 0.2, \ldots, 0.9\}$ for Storey-BH.
2. For each $\lambda \in R$, compute:

$$\widehat{\text{pFDR}}_\lambda(\gamma) = \frac{\hat{\pi}_0(\lambda)\gamma}{\widehat{\Pr}(p \leq \gamma)\{1 - (1 - \gamma)^m\}} \tag{6}$$

   where $\hat{\pi}_0(\lambda) = \hat{\pi}_0^{Storey}(\lambda)$ or $\hat{\pi}_0^{Quant}(\lambda)$ and $\widehat{\Pr}(p \leq \gamma)$ is the empirical estimate of $\Pr(p \leq \gamma)$

3. Generate $B$ bootstrap replicates $\{p_1^{*,b}, \ldots, p_m^{*,b}\}_{b=1}^B$ and compute $\widehat{\text{pFDR}}_\lambda^{*,b}(\gamma)$ for each $b$.
4. Estimate the MSE for each $\lambda$:

$$\widehat{\text{MSE}}(\lambda) = \frac{1}{B} \sum_{b=1}^B \left( \widehat{\text{pFDR}}_\lambda^{*,b}(\gamma) - \min_{\lambda' \in R} \widehat{\text{pFDR}}_{\lambda'}(\gamma) \right)^2 \tag{7}$$

5. Select $\hat{\lambda} = \arg\min_{\lambda \in R} \widehat{\text{MSE}}(\lambda)$

## F   IMPLEMENTATION DETAILS

**Experiment details.**  We run our experiments on NVIDIA GeForce RTX 4090 and NVIDIA L40 GPU, and implement all methods by PyTorch and vLLM.

**Dataset details.**   For the LLM QA datasets (MedMCQA, MMLU, and MMLU-Pro), we adopt the same prompts as in Luo et al. (2025). For the other datasets, we design our own prompts following the style of LLM QA prompts, as summarized in Table 3. In text labeling and LLM QA tasks, we merge the calibration and test sets to form the unlabeled dataset, except for MedMCQA. For MedMCQA, because test labels are unavailable, we use the calibration set as the unlabeled dataset. For image labeling, we use the validation sets of ImageNet and ImageNet-V2 as the unlabeled datasets.

## G   EXTENSION TO REGRESSION TASKS

While our primary focus is on classification, Conformal Labeling can be naturally extended to regression settings. Consider a loss function $L(Y, \hat{Y})$ that quantifies prediction error—for instance, the squared error $L(Y, \hat{Y}) = (Y - \hat{Y})^2$—alongside a user-specified tolerance level $\epsilon$. For each test sample, we define the null and alternative hypotheses as:

$$H_0^j : L(Y_{n+j}, \hat{Y}_{n+j}) > \epsilon \quad \text{versus} \quad H_1^j : L(Y_{n+j}, \hat{Y}_{n+j}) \leq \epsilon.$$

This framework generalizes the classification setting, which corresponds to the special case where $L(Y, \hat{Y}) = \mathbf{1}\{Y \neq \hat{Y}\}$ and $\epsilon = 0$. To apply Conformal Labeling in regression, we need an uncertainty score that reflects the model's predictive uncertainty. While uncertainty score functions

Table 3: Prompts for different datasets

| Dataset | Prompts |
|---|---|
| LLM QA | The following are multiple-choice questions. Give ONLY the correct option, no other words or explanation: |
| | [Question] A: [Option 1] B: [Option 2] C: [Option 3] D: [Option 4] Answer: [Mask] |
| Stance on global warming | You are given a statement about climate change. Determine the stance that a human would take towards this statement. |
| | Respond with ONLY the letter (A, B, or C) of the correct stance. Do not include any explanation. |
| | [Input Headline] A: agrees B: neutral C: disagrees Answer: [Mask] |
| Misinformation | You are a fact-checking assistant. Classify the following news headline as either real (A) or misinfo (B). |
| | Respond with ONLY the letter A or B. Do not include any explanation. |
| | Headline: [Input text] A: real B: misinfo Answer: [Mask] |

---

**Algorithm 2** Conformal Labeling Regression Tasks

---

**Require:** Calibration set $\mathcal{D}_{\text{cal}} = \{(X_i, Y_i)\}_{i=1}^{n}$; test instances $\{X_{n+j}\}_{j=1}^{m}$; pre-trained predictor $f$; loss function $L$; loss threshold $\epsilon$; target FDR level $\alpha \in (0, 1)$; nonconformity scores $\{S_i\}_{i=1}^{n+m}$

1: Identify calibration samples that exceed the loss threshold: $\mathcal{D}_{\text{cal}}^{0} = \{(X_i, Y_i) : L(Y_i, \hat{Y}_i) > \epsilon\}_{i=1}^{n}$, and set $n_0 = |\mathcal{D}_{\text{cal}}^{0}|$.

2: **for** $j = 1, \ldots, m$ **do**

3:     Compute the conformal $p$-value $\hat{p}_j$ according to equation 3.

4: **end for**

5: Apply the step-up procedure: $j^* = \max\left\{ j : \hat{p}_{(j)} \leq \frac{\alpha j (n+1)}{m(n_0+1)} \right\}$, where $\hat{p}_{(j)}$ is the $j$-th smallest $p$-value.

6: **Output:** The selected set $\mathcal{S} = \{j : \hat{p}_j \leq \hat{p}_{(j^*)}\}$.

---

are straightforward in classification tasks (e.g., $1 - \max_{y \in \mathcal{Y}} f_y(X)$), regression requires alternative approaches to get an uncertainty score function. For example, when using LLMs, we can leverage their verbalized confidence or prompt them to output prediction intervals, using the interval width as the uncertainty score. The complete procedure for Conformal Labeling in the regression task is outlined in Algorithm 2.

We evaluate Conformal Labeling on two regression problems: sentiment analysis with GPT-4o and protein structure prediction with AlphaFold. We use the data provided by Candès et al. (2025). For sentiment analysis, we prompt GPT-4o to output a prediction interval $[a_i, b_i]$ for each target $Y_i$, set the predicted value as $\hat{Y}_i = (a_i + b_i)/2$, and use the interval length $U_i = b_i - a_i$ as the uncertainty score. For protein structure prediction, we take experimentally derived structures as ground truth and AlphaFold predictions as $\hat{Y}_i$. Uncertainty scores are obtained from AlphaFold's internal confidence measure, the average predicted local distance difference test (pLDDT). For both experiments, we employ the L2 loss function.

In Table 4, we report the performance of Conformal Labeling on regression tasks with $\alpha = 0.1$. In all cases, Conformal Labeling consistently controls the realized FDR below the target level. A key observation is that Conformal Labeling's selection result in regression task is highly sensitive to the choice of the tolerance parameter $\epsilon$. For instance, in sentiment analysis, setting $\epsilon = 0.05$ yields an AI-labeled ratio of only $4.55\%$, whereas a slightly larger value $\epsilon = 0.06$ increases the ratio to $47.84\%$. In practice, selecting an appropriate $\epsilon$ may require domain expertise or prior knowledge about the task.

Table 4: **Performance of Conformal Labeling on regression tasks at $\alpha = 0.1$.** Results are shown for sentiment analysis (top) and protein folding (bottom) under different values of the tolerance parameter $\epsilon$. In all cases, Conformal Labeling controls the realized FDR below the target level.

| Dataset | Metric | Method | | |
|---------|--------|--------------------|--------------------|--------------------|
| | | CL ($\epsilon = 0.05$) | CL ($\epsilon = 0.06$) | CL ($\epsilon = 0.07$) |
| Sentiment analysis | FDR (%) | 8.48% | 8.60% | 7.89% |
| | Power (%) | 5.04% | 53.14% | 94.92% |
| | AI-labeled ratio | 4.55% | 47.84% | 85.43% |
| | | CL ($\epsilon = 1$) | CL ($\epsilon = 4$) | CL ($\epsilon = 9$) |
| Protein folding | FDR (%) | 9.67% | 9.90% | 9.04% |
| | Power (%) | 27.24% | 49.73% | 97.90% |
| | AI-labeled ratio | 10.06% | 36.80% | 88.00% |

Table 5: **Comparison of Conformal Labeling and Selective Guaranteed Risk (SGR) methods across different datasets and models.** Conformal Labeling achieves tight FDR control and high power, while SGR fails to control FDR.

| Dataset | Model | Method | FDR (%) | 90% Quantile FDP (%) | Power (%) |
|---------|-------|--------|---------|----------------------|-----------|
| ImageNet | ResNet34 | CL ($\alpha = 0.05$) | 4.99 | 5.65 | 63.99 |
| | | CL ($\alpha = 0.10$) | 9.90 | 10.55 | 79.75 |
| | | SGR ($\alpha = 0.05, \delta = 0.1$) | 3.88 | 4.40 | 57.89 |
| | | SGR ($\alpha = 0.10, \delta = 0.1$) | 8.73 | 9.45 | 76.86 |
| MMLU | Qwen3-32B | CL ($\alpha = 0.05$) | 4.90 | 6.04 | 61.31 |
| | | CL ($\alpha = 0.10$) | 9.96 | 11.10 | 82.81 |
| | | SGR ($\alpha = 0.05, \delta = 0.1$) | 6.39 | 4.68 | 16.10 |
| | | SGR ($\alpha = 0.10, \delta = 0.1$) | 7.71 | 8.76 | 76.62 |

# H  DISCUSSION

## H.1  DISCUSSION OF SELECTION WITH GUARANTEED RISK CONTROL.

Selection with Guaranteed Risk (SGR) (Geifman & El-Yaniv, 2017) is a framework designed to control prediction risk with high probability. Given a risk function $Risk$, SGR provides a high probability guarantee on the risk value. In our context, the risk function is defined as:

$$Risk = \frac{\mathbb{E}\left[\sum_{j=1}^{m} \mathbf{1}\{Y_{n+j} \neq \hat{Y}_{n+j} \text{ and } j \in \mathcal{R}\}\right]}{\mathbb{E}\left[\sum_{j=1}^{m} \mathbf{1}\{j \in \mathcal{R}\}\right]}.$$

For a given confidence parameter $\delta \geq 0$ and desired risk target $\alpha > 0$, SGR provides the guarantee:

$$\mathbb{P}\{Risk \leq \alpha\} \geq 1 - \delta.$$

This risk measure is also known as the marginal FDR (mFDR), which has been extensively compared with FDR in multiple testing literature. Notably, Benjamini & Hochberg (1995) favored FDR over mFDR because the latter cannot be controlled under the global null hypothesis. Storey (2003) also highlighted that mFDR fails to control the joint behavior between the number of false discoveries and the number of selected data.

To compare the performance of the two methods, we conduct experiments on MMLU and ImageNet datasets. In particular, we use maximum softmax probability (MSP) as the uncertainty score for both methods and run each experiment 100 times. The average and quantile results are presented in Table 5. The results show that our method can effectively achieve the target FDR while SGR fails (due to the target difference).

Table 6: **Performance of Conformal Labeling under varying severity levels of ImageNet-C Brightness corruption.** "No shift" denotes that the test dataset also comes from ImageNet. Conformal Labeling maintains stable FDR control close to target levels while power decreases with increasing corruption severity.

| Test Dataset | Accuracy (%) | Target FDR = 5% | | Target FDR = 10% | |
|---|---|---|---|---|---|
| | | FDR (%) | Power (%) | FDR (%) | Power (%) |
| No shift | 73.29 | 5.00 | 72.27 | 9.85 | 79.62 |
| Severity 1 | 68.98 | 5.88 | 65.08 | 11.74 | 81.36 |
| Severity 2 | 67.17 | 5.87 | 63.55 | 11.77 | 80.12 |
| Severity 3 | 64.15 | 5.94 | 61.52 | 11.70 | 78.05 |
| Severity 4 | 59.30 | 5.62 | 57.31 | 11.67 | 75.14 |
| Severity 5 | 52.75 | 5.56 | 53.94 | 11.75 | 72.09 |

## H.2 DISCUSSION OF DISTRIBUTION SHIFT

We emphasize that the calibration set is sampled i.i.d. from a large unlabeled dataset (a **transductive** setting), leaving the remaining data for testing. Given a large unlabeled dataset, we first annotate a small subset as the calibration set, then apply conformal labeling to guarantee the quality of AI labeling. This process ensures that the calibration and test sets are naturally i.i.d., satisfying the standard assumptions of conformal prediction. In other words, Conformal Labeling generally does not encounter distribution shift in practice.

While our method cannot provide theoretical guarantees under distribution shift, we add an experiment with ResNet34 on ImageNet and ImageNet-C Brightness to evaluate its empirical performance. In particular, we use ImageNet as the calibration set and ImageNet-C Brightness with varying severities as the testing set. We present the performance of our method across various testing sets in table 6. The results show that the realized power of our method is getting worse with a higher severity of distribution shift (may be due to the degraded accuracy), while the FDR is relatively insensitive. This demonstrates that our method is empirically robust to moderate distribution shift.

Besides, we note that prior work (Jin & Candès, 2025) has investigated approaches for handling covariate shifts in conformal novelty detection and conformal selection. The same idea can be naturally incorporated into our Conformal Labeling procedure, enabling it to maintain false discovery rate control under covariate shift. We believe it can be an interesting direction for subsequent works if there are some realistic scenarios with distribution shift.

# I DETAILED RESULTS

## I.1 COMPARISON TO SELECTIVE PREDICTION WITH CALIBRATED CONFIDENCES.

**Conformal Labeling achieves higher power than baselines under the same target FDR level.** To compare our method with selective prediction with calibrated confidences, we conduct a new experiment on ImageNet using a calibrated ResNet-34 classifier. In particular, we use a holdout dataset to learn an optimal temperature parameter for temperature scaling, achieving an ECE of $2.22\%$. Given an error level $\alpha$, we compare Conformal Labeling with two heuristic baselines. **(1) Confidence threshold.** Given a target error level $\alpha$, an instance is selected if its calibrated maximum softmax probability (MSP) satisfies $\mathcal{S}(x) > 1 - \alpha$. **(2) FDR search.** For a confidence threshold $t$, we define the empirical FDR on the calibration set as

$$\widehat{\mathrm{FDR}}(t) = \frac{\sum_{i=1}^{n} \mathbf{1}\{S(x_i) \geq t, \ Y_i \neq \hat{Y}_i\}}{\max\left\{\sum_{i=1}^{n} \mathbf{1}\{S(x_i) \geq t\}, \ 1\right\}}$$

We choose the largest threshold such that the empirical FDR does not exceed $\alpha$:

$$t^*(\alpha) = \max\{\, t : \widehat{\mathrm{FDR}}_{\mathrm{cal}}(t) \leq \alpha \,\}.$$

Then a test instance is selected if its MSP satisfies $\mathcal{S}(X_i) > t^*(\alpha)$.

Table 7: **Comparison of different selection methods at target error levels** $\alpha = 0.05$ **and** $\alpha = 0.10$**.** Conformal Labeling achieves the highest power and the tightest FDR control among the three methods.

| Method | $\alpha = 0.05$ | | $\alpha = 0.10$ | |
|---|---|---|---|---|
| | FDR (%) | Power (%) | FDR (%) | Power (%) |
| Confidence threshold | 2.80 | 57.23 | 3.94 | 66.60 |
| FDR search | 4.51 | 63.11 | 9.65 | 75.90 |
| Conformal Labeling | 5.00 | **72.27** | 9.97 | **79.91** |

Table 8: **Comparison of FDR control and power for MSP, Energy, and $D_\alpha$ score functions**. The experiment is conducted with a ResNet-34 model on ImageNet. **Bold** indicates the highest power under the target FDR level. All methods maintain FDR close to target levels, while MSP and $D_\alpha$ achieve significantly higher power than the Energy score.

| Method | Target FDR = 5% | | Target FDR = 10% | |
|---|---|---|---|---|
| | FDR (%) | Power (%) | FDR (%) | Power (%) |
| MSP | 4.97 | **63.87** | 9.97 | **80.01** |
| Energy | 4.86 | 11.83 | 9.94 | 50.58 |
| $D_\alpha$ | 4.95 | 63.80 | 9.92 | 79.74 |

We present the results of our method and two baselines in Table 7 below. The results show that Conformal Labeling consistently achieves the highest power and tightest FDR control among the three methods, showing that our method reliably increases the selected set size at the same target false discovery rate. Notably, we emphasize that the two heuristic baselines cannot provide any rigorous guarantee for the FDR control. We believe this demonstrates the benefits of our method compared to the baselines.

### I.2 COMPARISON OF DIFFERENT SCORE FUNCTIONS

**The choice of score function only affects the power but not the strict FDR control**   We present the results of three score functions on the ImageNet dataset in Table 8. While MSP, Energy, and $D_\alpha$ scores lead to vastly different power, all three successfully control the FDR below the target levels (5% and 10%). Specifically, MSP and $D_\alpha$ achieve high power (64% and 80% for $\alpha = 5\%$ and 10%, respectively), whereas Energy yields significantly lower power (11.83% and 50.58%). This demonstrates that a poor score function can incur higher data collection costs by lowering power, but it does not compromise the statistical guarantee of FDR control.

