# OpenReview forum: "Selective Labeling with False Discovery Rate Control"
_ICLR.cc/2026/Conference — Submitted to ICLR 2026_

### Official Review · Reviewer_UxdM · 2025-10-15

**Soundness:** 3
**Presentation:** 3
**Contribution:** 2
**Rating:** 4
**Confidence:** 3

**Summary:**

The paper addresses the problem of selecting a subset of reliable labeled examples from data labels produced by an AI  system. The paper proposes a variant of the FDR algorithm and provides a theoretical coverage guarantee.

**Strengths:**

The proposed method is very simple and easy to implement. It just sets a threshold on the prediction confidence and selects all the samples whose confidence is above this threshold.
The paper proves a theoretical coverage guarantee.

**Weaknesses:**

see questions

**Questions:**

Some of the datasets such as Imagenet have a large number of classes. There are uncertainty (conformity)  score functions tailored to large class sets such as Regularized Adaptive Prediction Sets (RAPS). It makes more sense to use them instead of MSP, which was used in the paper.

The proposed method seems to me as a variant of  FDR.  There is an ablation study in the paper that compares the proposed method to other variants of FDR. I think that these should be the baseline for comparison in Table 1. The difference between these FDR variants and the current method should be made clearer.

I think that an end-to-end experiment, which comprises label selection and using the selected samples to train a better system, can help to validate the effectiveness of the proposed system.

---

> ### Author Response · Authors · 2025-11-24
> **Response to Reviewer UxdM - part 1**
>
> Thanks for your valuable suggestions. Please find our response below:
>
> **1. Justification for the choice of uncertainty score [Q1]**
>
> Thank you for the comment. Uncertainty score functions like RAPS are designed for conformal prediction, which aims to construct **prediction sets** that contain the true label with a specific probability. However, we focus on selective labeling, where the goal is to select a subset where the model's **argmax predictions** are mostly correct. Therefore, score functions designed for conformal prediction (e.g., RAPS) can not be applied to our method.
>
> **2. Difference between FDR variants and the current method [Q2]**
>
> Thank you for the insightful comment. Indeed, our selection procedure can be treated as a variant of the Benjamini–Hochberg (BH) procedure. The BH procedure is known to be conservative by a factor $\pi_0$. The main difference is that prior works such as Storey’s correction [1] estimate $\pi_0$ from the **unlabeled test dataset** while Conformal Labeling estimates $\pi_0$ from the **labeled calibration dataset**. Following the suggestion, we present a detailed comparison between our method and variants of the BH procedure in Table 2 of Section 4.2. Here we present the results of different selection procedures for MedMCQA and MMLU datasets. The results show that **our method can achieve much higher power than those selection procedures, with more precise FDR control**.
>
>
> | Dataset | Model | Conformal Labeling FDR % | Conformal Labeling Power % | BH FDR % | BH Power %↑ | Storey-BH FDR % | Storey-BH Power %↑ | Quantile BH FDR % | Quantile BH Power %↑ |
> |---------|-------|--------------------------|----------------------------|----------|-------------|-----------------|-------------------|-------------------|---------------------|
> | MedMCQA | Llama-3.1-8B-Instruct | 9.70 | **31.44** | 3.68 | 12.89 | 7.43 | 25.37 | 6.33 | 22.11 |
> | MedMCQA | Qwen3-32B | 9.75 | **49.80** | 2.71 | 10.99 | 6.99 | 34.89 | 6.34 | 31.27 |
> | MedMCQA | Llama-3.1-70B-Instruct | 9.95 | **69.67** | 2.79 | 29.67 | 8.48 | 64.09 | 6.74 | 58.06 |
> | MMLU | Llama-3.1-8B-Instruct | 9.99 | **58.25** | 3.40 | 29.49 | 7.31 | 49.74 | 7.17 | 49.16 |
> | MMLU | Qwen3-32B | 10.00 | **82.96** | 1.37 | 10.51 | 7.33 | 74.57 | 6.25 | 70.37 |
> | MMLU | Llama-3.1-70B-Instruct | 9.96 | **88.20** | 1.61 | 18.74 | 6.95 | 80.08 | 5.96 | 76.77 | -->

---

> ### Author Response · Authors · 2025-11-24
> **Response to Reviewer UxdM - part 2**
>
> **3. An End-to-end experiment that  comprises label selection and using the selected samples to train a better system [Q3]**
>
> Thank you for the suggestion. To assess whether Conformal Labeling can help train a better system, we conduct an end-to-end experiment by training classifiers based on the selected samples. In particular, we generate 27000 synthetic data as a 10-class classification dataset, utilizing scikit-learn’s *make_classification*. We split the dataset into three parts: 2000 for training the labeling model ($D_1$), 20000 for AI labeling ($D_2$), 5000 for evaluation ($D_3$). First, we train a classifier on D1. Then, we use 2000 labeled data in D2 as calibration data and apply conformal labeling at target FDR $\alpha=0.1$ to the remaining 18000 unlabeled data, selecting a subset $D_{label}$ labeled by the classifier trained on $D_1$. The selection results for three different classifiers are reported in the table below.
>
> | Classifier | FDP (%) | Selection Size |
> |------------|-------|---------|
> | KNN (n=5)        | 9.72 | 6142   |
> | MLP (hidden_layer_sizes=(100, 25)) | 8.85 | 5715 |
> | XGBoost    | 9.56 | 4006   |
>
> After obtaining $D_{label}$, we use $D_1$ and $D_{label}$ to train a new classifier. In the table below, we present the prediction accuracy on $D_3$ for the classifier trained on $D_1$ and on $D_1 \cup D_{\text{label}}$. The results show that incorporating the selected samples into training consistently improves downstream accuracy. For example, when using KNN as the base classifier, training on $D_1 \cup D_{\text{label}}$ increases test accuracy from 64.92% to 69.60% (+4.68%). Similar improvements are observed for MLP (+2.60%) and XGBoost (+0.26%). Overall, these results indicate that Conformal Labeling enables the system to leverage unlabeled data to train a better model.
>
> | Classifier   | Training Set              | Accuracy (%)         | Improvement (%) |
> |--------------|---------------------------|----------------------|-------------|
> | KNN    | $D_1$                     | 64.92               | -           |
> |              | $D_1 \cup D_{\text{label}}$ | **69.60**  | **+4.68** ↑ |
> | MLP          | $D_1$                     | 65.68               | -           |
> |              | $D_1 \cup D_{\text{label}}$ | **68.28**  | **+2.60** ↑ |
> | XGBoost      | $D_1$                     | 59.80               | -           |
> |              | $D_1 \cup D_{\text{label}}$ | **60.06** | **+0.26** ↑     |
>
>
>
>
> **References**
>
> [1] Storey, John D. "A direct approach to false discovery rates." Journal of the Royal Statistical Society Series B: Statistical Methodology 64.3 (2002): 479-498. Oxford University Press.

---

### Official Review · Reviewer_CG4c · 2025-10-27

**Soundness:** 3
**Presentation:** 3
**Contribution:** 3
**Rating:** 6
**Confidence:** 3

**Summary:**

The paper introduces a procedure for safe automatic labeling: for each example, it either trusts the model’s prediction or defers to a human, while guaranteeing that the fraction of incorrect auto-labels stays below a chosen target level (FDR control).

**Strengths:**

Statistical guarantee: the method provides a finite-sample FDR guarantee (with proof provided in the appendix) on the fraction of incorrect labels among the accepted subset.

Applicability: the procedure treats the base model as a black box and only uses its confidence scores plus a small labeled calibration set. This makes it directly applicable to frozen large models without needing to train an abstention head, add a reject class, or modify the original architecture.

Experiments: the FDR guarantee is illustrated across diverse tasks and sensitivity experiments are provided.

**Weaknesses:**

Guarantee interpretation: the core guarantee controls false discovery rate in expectation over the randomness of the procedure, not per run. This makes it possible that a single run still contains an error rate above the target, which could be acknowledged more clearly in the paper (but this is not specific to this method in particular, but to CP based methods in general).

Additional baselines:  the experimental comparison could include additional baselines, such as abstention / selective prediction baselines (e.g. [1]).

[1] Geifman, Y., and El-Yaniv, R. “Selective Classification for Deep Neural Networks.” NeurIPS 2017.

**Questions:**

Could the authors provide additional baselines?

How to handle the case where the iid assumption or exchengability between the calibration and the test set does not hold?

---

> ### Author Response · Authors · 2025-11-24
> **Response to Reviewer CG4C - part 1**
>
> We appreciate the reviewer for the insightful and detailed comments. Please find our response below:
>
> **1. Guarantees interpretation [W1]**
> Thank you for raising this point. In the revised manuscript, we add the sentence "The realized FDR might exceed the target level for a single run." to clarify this point.
>
>
>
> **2. Discussion of Selection with guaranteed risk control [W2, Q1]**
>
> Thank you for the suggestion. We add new experiments by comparing our method to the new baseline -- Selection with Guaranteed Risk Control (SGR) [1]. It is worth noting that our method and SGR differ fundamentally in their theoretical guarantees: SGR provides a high-probability guarantees on the overall risk $\Pr\{R(f,g) > r^*\} < \delta$, while our method controls the false discovery rate (FDR), i.e., the expected proportion of incorrect predictions in the selected subset. The former is equivalent to the marginal FDR, which is less popular than FDR in the multiple testing literature. Thus, it is impractical to compare the performance of the two methods in a fair manner.
>
> Here, we present the results of our method and SGR across various hyperparameter settings to provide an informal comparison. In particular, we use maximum softmax probability (MSP) as the uncertainty score for both methods and run each experiment 100 times. The average and quantile results are presented in the table below. The results show that our method can effectively achieve the target FDR, while SGR fails (due to the target difference). We add this discussion in Appendix H.1 of the revised manuscript.
>
>
>
> | Dataset | Model | Method | FDR (%) | 90% quantile FDP (%) | Power (%) |
> |---------|-------|--------|---------|----------------------|-----------|
> | ImageNet | ResNet34 | CL ($\alpha$=0.05) | 4.99 | 5.65 | 63.99 |
> | |  | CL ($\alpha$=0.1) | 9.90 | 10.55 | 79.75 |
> | |  | SGR ($r^*$=0.05, $\delta$=0.1) | 3.88 | 4.40 | 57.89 |
> | |  | SGR ($r^*$=0.1, $\delta$=0.1) | 8.73 | 9.45 | 76.86 |
> | MMLU | Qwen3-32B | CL ($\alpha$=0.05) | 4.90 | 6.04 | 61.31 |
> | |  | CL ($\alpha$=0.1) | 9.96 | 11.10 | 82.81 |
> | |  | SGR ($r^*$=0.05, $\delta$=0.1) | 6.39 | 4.68 | 16.10 |
> | |  | SGR ($r^*$=0.1, $\delta$=0.1) | 7.71 | 8.76 | 76.62 |

---

> > ### Author Response · Authors · 2025-11-24
> > **Response to Reviewer CG4C - part 2**
> >
> > **3. How to handle distribution shift? [Q2]**
> >
> > Thank you for the suggestion. Before presenting the empirical results, we emphasize that the calibration set is sampled i.i.d. from a large unlabeled dataset (a **transductive** setting), leaving the remaining data for testing. Given a large unlabeled dataset, we first annotate a small subset as the calibration set, then apply conformal labeling to guarantee the quality of AI labeling. This process ensures that the calibration and test sets are naturally i.i.d., satisfying the standard assumptions of conformal prediction. While our method cannot provide theoretical guarantees under distribution shift, we add a new experiment with ResNet34 to evaluate its empirical performance as the reviewer suggested. In particular, we use ImageNet as the calibration set and ImageNet-C with varying severities as the testing set. We present the performance of our method across various testing sets in the table below. The results show that **the realized power of our method is getting worse with a higher severity of distribution shift** (may be due to the degraded accuracy), while **the FDR is relatively insensitive**. This demonstrates that our method is empirically robust to moderate distribution shift.
> >
> > Besides, we note that prior work [2] has investigated approaches for handling covariate shifts in conformal novelty detection and conformal selection. The same idea can be naturally incorporated into our Conformal Labeling procedure, enabling it to maintain false discovery rate control under covariate shift. We believe it can be an interesting direction for subsequent works if there are some realistic scenarios with distribution shift. We add the analysis in Appendix H.1 of the revised manuscript.
> >
> >
> >
> > | Dataset | Model | Conformal Labeling |        | BH |        | Storey-BH |        | Quantile-BH |        |
> > |---------|--------|--------------------|--------|----|--------|-----------|--------|-------------|--------|
> > |         |        | FDR (%) | Power (%)     | FDR (%) | Power (%) | FDR (%) | Power (%) | FDR (%) | Power (%) |
> > | MedMCQA | Llama-3.1-8B-Instruct      | 9.70 | **31.44** | 3.68 | 12.89 | 7.43 | 25.37 | 6.33 | 22.11 |
> > | MedMCQA | Qwen-3-32B                 | 9.75 | **49.80** | 2.71 | 10.99 | 6.99 | 34.89 | 6.34 | 31.27 |
> > | MedMCQA | Llama-3.1-70B-Instruct     | 9.95 | **69.67** | 2.79 | 29.67 | 8.48 | 64.09 | 6.74 | 58.06 |
> > | MMLU    | Llama-3.1-8B-Instruct      | 9.99 | **58.25** | 3.40 | 29.49 | 7.31 | 49.74 | 7.17 | 49.16 |
> > | MMLU    | Qwen-3-32B                 | 10.00 | **82.96** | 1.37 | 10.51 | 7.33 | 74.57 | 6.25 | 70.37 |
> > | MMLU    | Llama-3.1-70B-Instruct     | 9.96 | **88.20** | 1.61 | 18.74 | 6.95 | 80.08 | 5.96 | 76.77 |
> >
> >
> >
> > **References**
> >
> > [1] Geifman, Y., and El-Yaniv, R. “Selective Classification for Deep Neural Networks.” NeurIPS 2017.
> >
> > [2] Jin, Ying, and Emmanuel J. Candès. "Model-free selective inference under covariate shift via weighted conformal p-values." Biometrika (2025): asaf066. Oxford University Press.

---

### Official Review · Reviewer_ofxs · 2025-11-01

**Soundness:** 3
**Presentation:** 3
**Contribution:** 2
**Rating:** 6
**Confidence:** 4

**Summary:**

The paper proposes a way to use an AI model to label part of a dataset while keeping the fraction of wrong AI labels under control. The method is called Conformal Labeling. It treats each test example as a hypothesis test. It compares the model’s confidence on that example with confidences observed on a small calibration set where the model is known to be wrong. This produces a p value per example. The method then selects the largest set of examples whose p values pass a data driven threshold, giving a guarantee that the false discovery rate stays below a user chosen level. The authors prove the guarantee and show experiments on image classification, text labeling, and multiple choice question answering with language models. They report tight false discovery rate control and strong coverage. For example, on ImageNet with ResNet 34 they can let the model label about fifty nine percent of the data while keeping the false discovery rate below ten percent.

**Strengths:**

Framing selective labeling as multiple testing with conformal p values is a clean and useful idea. The paper is not just another confidence thresholding scheme. It borrows mature tools from conformal inference and false discovery rate control, and adapts them to this labeling setting.

There is a clear theorem that states the procedure controls false discovery rate at the chosen level, under standard iid sampling of calibration and test points. The selection rule resembles Benjamini–Hochberg but is modified to account for the number of misclassified calibration examples. The paper also studies the choice of uncertainty score and shows that maximum softmax probability tends to separate correct and incorrect predictions better than the energy score in their setup.

The narrative is easy to follow. The three steps are named clearly: compute uncertainty scores, compute conformal p values, and apply thresholding. The method section states the null and alternative in plain terms

Selective labeling is widely needed in practice when annotation budgets are tight. Prior work either used heuristics or guaranteed only the overall dataset error by relying on humans to cancel model mistakes. By guaranteeing the quality of the AI labeled subset itself, this work addresses a real pain point.

**Weaknesses:**

The key guarantee assumes that calibration and test come from the same distribution and that the calibration set identifies a sufficient number of model mistakes. In real pipelines, shifts are common and the model may be updated after calibration. The paper would benefit from stress tests with shift between calibration and test, and from guidance on how often to refresh calibration in a streaming setting.

The power analyses show that maximum softmax probability and the alpha score work well, while the energy score does not separate as nicely. This means the method inherits weaknesses of the uncertainty score. The paper adopts maximum softmax probability for the main results because it is simple.

The baselines are a naive score threshold and labeling everything with the model. These are useful, but the paper would feel stronger with comparisons to more advanced selective prediction or abstention systems that use calibrated confidence, temperature scaling, or selective risk minimization, even if those do not provide the same guarantee.

For multiple choice QA, the method treats answer options as classes and takes logits for option tokens. This is standard, but many modern evaluations use chain of thought or candidates longer than a single token. The method should clarify how to handle multi token options and prompt sensitivity.

**Questions:**

How sensitive is the guarantee and the realized power to moderate shift between calibration and test?

Have you tried training a small auxiliary model to predict correctness on the calibration set, and then using that learned score in place of maximum softmax probability?

If I already run a selective prediction model that abstains based on calibrated confidence, what concrete benefit do I get by wrapping your conformal procedure around it? Does your threshold reliably increase the selected set size at the same target false discovery rate, and can you show this on at least one dataset in the main results instead of only in the ablation figure?

---

> ### Author Response · Authors · 2025-11-24
> **Response to Reviewer ofxs - part 1**
>
> Thank you for the valuable comments and detailed feedback. Please find our response below:
>
> **1. Performance under moderate distribution shift [W1, Q1]**
>
> Thank you for the suggestion. Before presenting the empirical results, we emphasize that the calibration set is sampled i.i.d. from a large unlabeled dataset (a **transductive** setting), leaving the remaining data for testing. Given a large unlabeled dataset, we first annotate a small subset as the calibration set, then apply conformal labeling to guarantee the quality of AI labeling. This process ensures that the calibration and test sets are naturally i.i.d., satisfying the standard assumptions of conformal prediction. While our method cannot provide theoretical guanranree under distribution shift, we add a new experiment with ResNet34 to evaluate its empirical performance as the reviewer suggested. In particular, we use ImageNet as the calibration set and ImageNet-C Brightness with varying severities as the testing set.
> We present the performance of our method across various testing sets in the table below. The results show that **the realized power of our method is getting worse with a higher severity of distribution shift** (may due to the degraded accuracy), while **the FDR is relatively insensitive**. This demonstrates that our method is empirically robust to moderate distribution shift. Besides, for the online setting, one may use a variant of our method by employing the online Benjamini–Hochberg procedure [1], which performs real-time selection while providing FDR control. We also add the analysis in Appendix H.2 of the revised manuscript.
>
>
>
> | Test Dataset                          | Accuracy (%) | Target FDR = 5%          |                  | Target FDR = 10%         |                  |
> |---------------------------------------|--------------------|--------------------------|------------------|--------------------------|------------------|
> |                                       |                    | Realized FDR             | Power (%)        | Realized FDR             | Power (%)        |
> | ImageNet (no shift)           | 73.29          | 4.86                | 63.47        | 9.85                | 79.62        |
> | ImageNet-C Brightness (severity 1)    | 68.98              | 5.88                    | 65.08            | 11.74                   | 81.36            |
> | ImageNet-C Brightness (severity 2)    | 67.17              | 5.87                    | 63.55            | 11.77                   | 80.12            |
> | ImageNet-C Brightness (severity 3)    | 64.15              | 5.94                    | 61.52            | 11.70                   | 78.05            |
> | ImageNet-C Brightness (severity 4)    | 59.30              | 5.62                    | 57.31            | 11.67                   | 75.14            |
> | ImageNet-C Brightness (severity 5)| 52.75          | 5.56                | 53.94        | 11.75               | 72.09        |
>
> **2. Dependence on the choice of uncertainty score [W2]**
>
> Thank you for raising this concern. We clarify that the choice of uncertainty score does not influence the guarantee of FDR control. Instead, a bad uncertainty score can lead to a low power in our method, i.e., a low proportion of reliable AI labeling. In other words, it does not violate the validity of our method but can increase the costs by requiring more collected data. This is demonstrated by the following table, which shows the realized FDR and power of different score functions for a ResNet-34 model on ImageNet. We also add this experiment in Appendix I.2 of the revised manuscript.
>
> | Score | Target FDR = 5% | | Target FDR = 10% | |
> |:-------|:---------------:|:---------------:|:----------------:|:----------------:|
> | | FDR (%) | Power (%) | FDR (%) | Power (%) |
> | **MSP** | 4.97 | 63.87 | 9.97 | 80.01 |
> | **Energy** | 4.86 | 11.83 | 9.94 | 50.58 |
> | **$D_\alpha$** | 4.95 | 63.80 | 9.92 | 79.74 |

---

> ### Author Response · Authors · 2025-11-24
> **Response to Reviewer oxfs - part 2**
>
> **3. How to handle multi-token options and how sensitive is the method to prompts [W4]**
>
> Thank you for raising this important point.
> > How to handle multi-token options
>
> The challenge in applying our method to multi-token question answering lies in designing an effective confidence measure, which has been well studied in the LLM community. For example, we can use the P(True) formulation [2] by asking the model a binary “Yes/No” question regarding whether its own generated answer is correct, or we can use verbalized confidence of the model.
>
> >  Prompt sensitivity
>
> To evaluate the prompt sensitivity of our method, we conducted experiments with Qwen3-32B on the MMLU dataset using three distinct prompts. The confidence measure is obtained using logits-based approach. The prompts are defined as follows:
>
> • **Prompt 1**
> `"The following are multi choice questions. Give ONLY the correct option, no other words or explanation:\nQuestion: {question}\nA: {choice1}\nB: {choice2}\nC: {choice3}\nD: {choice4}\nAnswer: "`
>
> • **Prompt 2**
> `"You will be given multiple-choice questions. Respond with ONLY the letter of the correct choice. No explanations.\n\nQuestion: {question}\n\nA: {choice1}\nB: {choice2}\nC: {choice3}\nD: {choice4}\n\nAnswer: "`
>
> • **Prompt 3**
> `"Answer the following multiple-choice question. Output ONLY the correct option (A, B, C, etc.). No other text.\n\nQuestion: {question}\n\nOptions:\nA: {choice1}\nB: {choice2}\nC: {choice3}\nD: {choice4}\n\nCorrect option: "`
>
> The table below presents the performance of our method with different prompts. The results show that the FDR guarantee of our method is insensitive to the prompt design. Besides, the prompt 3 leads to lower power of our method than the other two prompts. This is because the LLM achieves a higher testing error with this prompt. In summary, our method can be employed by LLM with different prompts.
>
>
> | Prompt | Mean FDR (%) | Mean Power (%) | Error (%)   |
> |--------|--------------|----------------|-------------|
> | 1      | 9.97         | 82.99          | 21.46       |
> | 2      | 9.94         | 82.58          | 21.48       |
> | 3      | 10.00        | 79.18          | 22.10       |
>
> **4. Try training a small auxiliary model to predict correctness on the calibration set, and then using that learned score in place of maximum softmax probability [Q2]**
>
> Thank you for the question. We conduct a new experiment by learning a small MLP to predict the correctness in the calibration set. Note that training on calibration data would violate the i.i.d. assumption and invalidate the theoretical guarantees, we therefore train the predictor on a separate hold-out set of 5,000 samples using a small MLP (512→64 ReLU→2) with cross-entropy loss, taking ResNet-34 features as input to predict prediction correctness. We present the empirical results of learned score and MSP in the table below. The results show that the learned score achieves much lower power than MSP, with poor performance in ECE [3] and accuracy. This demonstrates that predicting the correctness cannot provide a reliable metric for data selection or confidence estimation.
>
>
> | Score           | FDR (%) | Power (%) | ECE (%) | correctness-prediction accuracy (%) |
> |-----------------|---------|-----------|---------|-----------------------------------|
> | Learned Score   |  9.91       | 10.43     | 22.22   | 65.01                            |
> | MSP             |    9.97     | 80.01     | 3.78    | —                                 |

---

> ### Author Response · Authors · 2025-11-24
> **Response to Reviewer oxfs - part 3**
>
> **5. Comparison to selective prediction with calibrated confidences. [W3, Q3]**
>
> Thank you for the insightful suggestion. To compare our method with selective prediction with calibrated confidences, we conduct a new experiment on ImageNet using a calibrated ResNet-34 classifier. In particular, we use a holdout dataset to learn an optimal temperature parameter for temperature scaling, achieving an ECE of $2.22\%$. Given an error level $\alpha$, we compare Conformal Labeling with two heuristic baselines: (1) **Confidence threshold**: we abstain from prediction when the calibrated maximum softmax probability (MSP) $\leq1-\alpha$; (2) **FDR search**: we select the largest threshold such that the FDR on the calibration set is smaller than $\alpha$ and abstain from prediction when the MSP is smaller than the threshold.
>
> We present the results of our method and two baselines in the table below. The results show that Conformal Labeling consistently achieves the highest power and tightest FDR control among the three methods, showing that our method **reliably increases the selected set size** at the same target false discovery rate. Notably, we emphasize that the two heuristic baselines cannot provide any rigorous guarantee for the FDR control. We believe this demonstrates the benefits of our method compared to the suggested baselines. We add the experiment in Appendix I.1 in the revised manuscript.
>
>
>
> | Method | FDR (%) | Power (%) | FDR (%) | Power (%) |
> |:---|:---:|:---:|:---:|:---:|
> | Target error level | $\alpha=0.05$ | | $\alpha=0.10$ | |
> | Confidence threshold | 2.80 | 57.23 | 3.94 | 66.60 |
> | FDR search| 4.51 | 63.11 | 9.65 | 75.90 |
> | Conformal Labeling | 5.00 | **72.27** | 9.97 | **79.91** |
>
>
>
> **References**
>
> [1] Fischer, Lasse, Ziyu Xu, and Aaditya Ramdas. "Online generalizations of the e-BH and BH procedure." arXiv preprint arXiv:2407.20683 (2024).
>
> [2] Shen, M., Das, S., Greenewald, K., Sattigeri, P., Wornell, G. W., and Ghosh, S. "Thermometer: Towards Universal Calibration for Large Language Models." ICML 2024.
>
> [3] Naeini, M. P., Cooper, G., and Hauskrecht, M. "Obtaining Well Calibrated Probabilities Using Bayesian Binning." AAAI 2015.

---

### Official Review · Reviewer_WZEv · 2025-11-01

**Soundness:** 3
**Presentation:** 2
**Contribution:** 2
**Rating:** 2
**Confidence:** 3

**Summary:**

The authors propose a new method for assuring the quality of AI model-based labeling of datasets. The method aims to find the maximum-size subset of the dataset where AI's errors (false discovery rate; FDR) can be below a given rate in expectation, while maximizing ratio of correct labels. The paper proposes to compute a p-score based on a score function (i.e. AI model's prediction confidence), which is then thresholded to decide what samples to allocate to the model for annotation: the authors provide marginal guarantee for the FDR to be under a certain level. The authors test their method with different datasets/tasks and models, as well as under various ablations.

Overall I have to recommend the rejection of the paper as in my understanding 1- the utility of what they are trying to achieve is not very clear (see weaknesses, pt. 1), and 2- they do not deliver on these claims fully, at least in comparison to their presentation in the abstract and the introduction (see weaknesses, pt. 2). I cannot recommend a borderline rejection since these issues are core problems rather than superficial ones. However, if the review scale allowed I would have assigned this paper a 3, since I believe the overall direction of the paper is important, and their approach has some merit.

**Strengths:**

- Given the increasing need for AI-based annotation of datasets, the topic is very timely and important.
- The authors overall motivation is valid: being able to provide guarantees for subsets of a dataset is an important target, which could potentially offer utility that is qualitatively different than the dataset-wise guarantees (but see below)
- The experiments contain a good variety in datasets and models for testing the method.
- The authors conduct ablation experiments with score functions, calibration set sizes, and threshold selection method which is very helpful for a conformal inference-based method.

**Weaknesses:**

- I think the motivation of the paper is not very clear. The paper states that "This limitation underscores a critical gap: existing selective labeling methods cannot ensure the quality of AI-assigned labels, hindering their reliable deployment in real-world applications." Why would be separately interested in AI-labeled instances' reliability - isn't the overall dataset label reliability the fundamental objective? I can understand this reasoning if the current method was improving overall performance by utilizing a different source for the score function/heuristic than model certainty to determine annotator allocation (human vs. AI). However this is not the case as both papers are using AI model confidence, as stated in L151 for the current paper.
- Moreover, it seems like the paper underdelivers relative to its claims. Although their procedure controls the FDR of the subset selected, it controls it *in expectation*. So, the authors' criticism of Candes et al. 2025 that "the labeling error of AI models can be unacceptably high, even reaching 100%" (L048) could apply to their own method as well. In fact, the high-probability, dataset-wise guarantee provided by the prior work can be considered preferable as it is a sample-specific guarantee. Again, although the explicit control over FDR vs. power provided by the current work is not unimportant, this can be emulated by Candes et al. 2025's method by increasing the human-annotated instance set size in the test set. If I am correct about the nature of the guarantee provided by the paper, then claims like "In this work, we study the problem of identifying instances where AI predictions can be provably trusted." (L090) are not substantiated. Although some experimental findings are helpful (L364), I think it is not appropriate to relegate the discussion and addressing of this core issue to empirical experiments.

**Questions:**

- Please fix the page-long references in the bibliography
- L047: I do not exactly understand this criticism. In selective labeling the reasonable assumption is that a considerable portion of the dataset will be annotated by AI. Is there a realistic scenario under PAC labeling where AI error is 100%?
- L054: I think this part is hard to understand without the technical prelude. So the authors can go for a more verbalistic summary or provide a little more technical background
- L077: FDR control has already been discussed, it reads redundant for it to be repeated in every bullet point in the contributions.
- L098: What does "Since AI models are typically prone to labeling errors" mean? I.e. do they make errors when used for labeling or are they affected by labeling errors?
- L102: What is the expectation over? Please explicitly state. Also goes for L111.
- L108: "labeling as many test instances as possible *correctly*"
- L189: $\mathcal{S}$ is treated both as a function (L155) and a random variable without any acknowledgment
- L191: Why call the LHS variable $\hat{p}\_j$ instead of $\hat{p}\_{n+j}$?
- L216: I agree with the authors' claim that energy score is worse; but y-axes sharing among the figures can make this more obvious
- L300: I feel by this point conformal labeling can be abbreviated (CL)
- L459: Redundant period before citations.
- L465: Reference to the selective prediction methods is a nice touch

---

> ### Author Response · Authors · 2025-11-24
> **Response to Reviewer WZEv - part 1**
>
> We sincerely appreciate the reviewer’s constructive feedback. Please find our response below:
>
> **1. Motivation is not clear [W1]**
>
> Thank you for pointing out the ambiguous presentation. We recognize that providing no guarantee on AI-assigned labels is not a "limitation" of PAC labeling but different targets from our method. PAC labeling excels at guaranteeing the overall dataset error, while our method, Conformal Labeling, is designed to guarantee the error within the AI-labeled subset. In other words, the two approaches are designed for different scenarios. In the revised manuscript (see 2nd paragraph of Introduction), we revise the motivation part and provide concrete examples where a reliable AI-labeled subset is critical, as presented below.
>
>
> > "Why would be separately interested in AI-labeled instances' reliability?"
>
> We argue that controlling reliability at the overall dataset level can be impractical in large-scale applications. For instance, given a massive unlabeled dataset (e.g., LAION-5B), practitioners typically annotate only a subset via AI labeling, leaving the remaining instances without human annotation (due to its large-scale nature). Additionally, one might use LLMs to synthesize extensive domain-specific texts, followed by another AI model to label a portion of them with guarantees on error rates (such as FDR control). By iterating this synthesis-and-labeling process, it becomes feasible to build a vast annotated dataset with bounded labeling errors. Thus, it is critical to focus reliability efforts specifically on the AI-labeled subset in such scenarios.
>
>
>
> **2. Justification to Overclaims [W2]**
>
> Thank you for pointing out this problem. We response to the three points separately.
> > "the labeling error of AI models can be unacceptably high, even reaching 100%" (L048) could apply to their own method as well
>
> Yes, our method also only controls the error in expectation, so the error can be 100% with low probability. However, we clarify that this sentence is to show the labeling error of AI models in PAC labeling can be unacceptably high even in expectation, leaving a large proportion of instances to human annotation. In the revised manuscript, we remove this sentence to avoid any misunderstanding.
>
> > Although the explicit control over FDR vs. power provided by the current work is not unimportant, this can be emulated by Candes et al. 2025's method by increasing the human-annotated instance set size in the test set.
>
> Indeed, PAC labeling can achieve similar FDR with a refined hyperparameter through grid search. However, it cannot directly control the FDR via explicitly setting it in the hyperparameter value. Intuitively, one can also simply search the threshold to achieve the FDR similar to PAC labeling. The problem is that such an approach only achieves the FDR vs. power trade-off heuristically, without a theoretical guarantee on FDR. In contrast, our method is explicitly designed to maximize power with FDR guarantee, a target that existing methods cannot achieve rigorously.
>
> > Claims like "In this work, we study the problem of identifying instances where AI predictions can be provably trusted." are not substantiated.
>
> In this work, we propose a rigorous method to control the FDR of AI annotations, i.e., the expected fraction of incorrect AI predictions among the selected subset. In other words, our method can identify subsets where AI predictions can be provably trusted. To avoid any ambiguity, we replace the "identifying instances" with "identifying subsets" in the revised manuscript.

---

> ### Author Response · Authors · 2025-11-24
> **Response to Reviewer WZEv - part 2**
>
> **3. Concerns of presentation [Q1, Q3, Q5, Q6, Q7, Q8, Q10, Q12]**
>
> We thank the reviewers for identifying these problems and providing practical suggestions. The following corrections have been made in the revised manuscript:
>
> • *page-long reference [Q1]*: The lengthy reference has been shortened.
>
> • *L054 hard to understand [Q3]*: We add a sentence "The $p$-values quantify how abnormal the test prediction is compared with incorrect predictions in the calibration dataset." after introducing the construction of our $p$-values to help readers understand the usage of the $p$-values.
>
> • *Meaning of "Since AI models are typically prone to labeling errors" [Q5]*: It means they make errors when used for labeling. To enhance clarity, we modify it into "Since AI models unavoidably make errors when used for labeling".
>
> • *What is the expectation over? [Q6]*: The expectation is over the randomness of both the calibration dataset and test dataset. In the revised manuscript, we clarify this point. At line TBD, we specify that the expectation is taken over the data's randomness. In Theorem 3.1, we specify that the expectation is over both the calibration and test data.
>
> • *"labeling as many test instances as possible correctly"* [Q7]: The adverb "correctly" is added.
>
> • *$\mathcal{S}$ is treated both as a function and a random variable without any acknowledgment [Q8]*: In L 199, we define
> $ \mathcal{S}_i = 1 - \text{max}_y f_y(X_i)$
>  before we use $ \mathcal{S}_i $ as variables.
>
> • *Y-axis sharing for Figure 1 [Q10]*: We share y-axes among the figures to make the comparisons more obvious.
>
> • *Redundant period before citations. [Q12]*: The redundant period is removed.
>
> **4. Is there a realistic scenario under PAC labeling where AI error is 100%? [Q2]**
>
> We clarify that this sentence is to present an extreme case: PAC labeling can control the overall error even when the error of AI labeling is 100%. This is because PAC labeling can increase the proportion of human annotations to achieve the target error, despite the increased costs. We agree that this is not a limitation of PAC labeling, so we remove this claim in the revised manuscript.
>
> **5. Repetitive discussion of FDR [Q4]**
>
> Thank you for the feedback. We think this repetition is necessary to give readers a clear understanding of what each step of Conformal Labeling is designed to achieve.
>
> **7. Why using $p_{j}$ instead of $p_{n+j}$ [Q9]**
>
> Thank you for the comment. We use $p_{j}$ instead of $p_{n+j}$ because we only construct $p$-values for the test data and do not construct $p$-values for the calibration data. Besides, prior works [1, 2] also use the notion $p_{j}$ instead of $p_{n+j}$.
>
> **8. Conformal labeling could be abbreviated [Q11]**
>
> Thank you for the suggestion. We prefer to keep the full name.
>
> **9. Reference to the selective prediction methods [Q13]**
>
> Thank you for the suggestion. In the related work section, we add references to prior selective prediction methods.
>
> **References**
>
> [1] Jin, Ying, and Emmanuel J. Candès. "Selection by prediction with conformal p-values." Journal of Machine Learning Research, vol. 24, no. 244, 2023, pp. 1–41.
>
> [2] Bai, Tian, Yue Zhao, Xiang Yu, and Archer Y. Yang. "Multivariate Conformal Selection." In Forty-second International Conference on Machine Learning, 2025.

---

### Author Response · Authors · 2025-11-24
**General Response**

We appreciate the reviewers’ thoughtful feedback and valuable comments on our work. We are encouraged that reviewers recognize the problem of selective labeling as both **timely and practically important** (WZEv, oxfs), and highlight that framing selective labeling as a multiple-testing problem with conformal p-values is a **clean and useful** idea (oxfs). We are also pleased that reviewers find our method **simple**, **intuitive**, and **easy to implement** (oxfs, UxdM), while also being **theoretically grounded** (oxfs, CG4c). In addition, we are grateful for the acknowledgment of our **extensive experimental evaluation** (WZEv), which includes ablations on uncertainty score functions, thresholding strategies, and other key design choices.


In the following responses, we have addressed the reviewers' comments and concerns point by point. We have also revised the manuscript according to reviewers' suggestions, and we believe this makes our paper much stronger. The main changes$^1$ we made are summarized below:
* Revised claims in **Abstract** [WZEv].
* Clarified the motivation of Conformal Labeling and its relation to prior work in the second paragraph of **Introduction** and the second paragraph of **Prelimiaries**. [WZEv]
* Revised the third paragraph of **Introduction** to avoid repetition of FDR. [WZEv]
* Clarified that realized FDR might exceed the target level for a single run as a footnote in **Page 3**. [CG4c]
* Clarified what the expectation of FDR is over in **Line 107** and **Theorem 3.1**. [WZEv]
* Added comparisons between Conformal Labeling's selection procedure and other selection procedures in **section 4.2**. [UxdM]
* Added discussion on selective prediction and conformal inference for classification in **Related Work**. [UxdM]
* Revised the proof of Theorem 3.1 in **Appendix B**.
* Added discussion on the difference between Conformal Labeling and prior selective prediction methods in **Appendix H.1**.
* Added discussion on distribution shift in **Appendix H.2**. [oxfs, CG4c]
* Added comparisons with selective prediction with calibrated confidences in **Appendix I.1**. [oxfs]
* Added more detailed comparisons of score functions in **Appendix I.2**. [oxfs]

$^1$ For clarity, we highlight the revised part of the manuscript in **blue** color.

---

### Meta-Review · Area_Chair_wEEk · 2026-01-07

**Summary:**

Reviewer **WZEv** recognizes that the topic of this paper is very important, but with the main concern: the motivation of this paper is not clear, and the overclaim issue.

Reviewer **ofxs** recognizes the usefulness of the idea. The main concerns include the distribution shift issue between the calibration set and the test set, and the lack of comparison with more advanced selective prediction methods. The authors added an experiment to evaluate their method under distribution shift and to prove its insensitivity. They also compared with more selective prediction methods.

Reviewer **CG4c** wants the authors to add more baselines and also has concerns about the non-iid situation. The authors gave a good rebuttal to those two questions.

Reviewer **UxdM** has concerns about the use of RAPS in this paper.

**Reviewer Concerns:**

I think Reviewer **ofxs**, **CG4c**, and **UxdM**’s concerns have been well solved. But the Reviewer **WZEv**’s major concerns remain.

**Reviewer Scores:**

I think Reviewer **UxdM** may raise his score, but Reviewer **WZEv** will keep the negative rating.

---

### Decision · Program_Chairs · 2026-01-26

Reject